# Quantum Wasserstein GANs

**Shouvanik Chakrabarti**[1,2,4,*] **Yiming Huang**[3,1,5,*] **Tongyang Li**[1,2,4]
**Soheil Feizi**[2,4] **, Xiaodi Wu**[1,2,4]

[1] Joint Center for Quantum Information and Computer Science, University of Maryland

[2] Department of Computer Science, University of Maryland

[3] School of Information and Software Engineering

University of Electronic Science and Technology of China

[4] {shouv,tongyang,sfeizi,xwu}@cs.umd.edu

[5] yiminghwang@gmail.com

## Abstract

The study of quantum generative models is well motivated, not only because of its importance in quantum machine learning and quantum chemistry but also because of the perspective of its implementation on near-term quantum machines. Inspired by previous studies on the adversarial training of classical and quantum generative models, we propose the first design of quantum Wasserstein Generative Adversarial Networks (WGANs), which has been shown to improve the robustness and the scalability of the adversarial training of quantum generative models even on noisy quantum hardware. Specifically, we propose a definition of the Wasserstein semimetric between quantum data, which inherits a few key theoretical merits of its classical counterpart. We also demonstrate how to turn the quantum Wasserstein semimetric into a concrete design of quantum WGANs that can be efficiently implemented on quantum machines. Our numerical study, via classical simulation of quantum systems, shows the more robust and scalable numerical performance of our quantum WGANs over other quantum GAN proposals. As a surprising application, our quantum WGAN has been used to generate a 3-qubit quantum circuit of ∼50 gates that well approximates a 3-qubit 1-d Hamiltonian simulation circuit that requires over 10k gates using standard techniques.

## 1 Introduction

Generative adversarial networks (GANs) [19] represent a power tool of training deep *generative* models, which have a profound impact on machine learning. In GANs, a generator tries to generate fake samples resembling the true data, while a discriminator tries to discriminate between the true and the fake data. The learning process for generator and discriminator can be deemed as an adversarial game that converges to some equilibrium point under reasonable assumptions.

Inspired by the success of GANs and classical generative models, developing their quantum counterparts is a natural and important topic in the emerging field of quantum machine learning [5, 37]. There are at least two appealing reasons for which quantum GANs are extremely interesting. First, quantum GANs could provide potential quantum speedups due to the fact that quantum generators and discriminators (i.e., parameterized quantum circuits) cannot be efficiently simulated by classical generators/discriminators. In other words, there might exist distributions that can be efficiently generated by quantum GANs, while otherwise impossible with classical GANs. Second, simple prototypes of quantum GANs (i.e., executing simple parameterized quantum circuits), similar to those of the variational methods (e.g., [16, 27, 30]), are likely to be implementable on near-term noisy-intermediate-scale-quantum (NISQ) machines [33]. Since the seminal work of [25], there are

---

quite a few proposals (e.g, [4, 13, 23, 34, 39, 46, 49]) of constructions of quantum GANs on how to encode quantum or classical data into this framework. Furthermore, [23, 49] also demonstrated proof-of-principle implementations of small-scale quantum GANs on actual quantum machines.

A lot of existing quantum GANs focus on using quantum generators to generate classical distributions. For truly quantum applications such as investigation of quantum systems in condensed matter physics or quantum chemistry, the ability to generate *quantum data* is also important. In contrast to the case of classical distributions, where the loss function measuring the difference between the real and the fake distributions can be borrowed directly from the classical GANs, the design of the loss function between real and fake quantum data as well as the efficient training of the corresponding GAN is much more challenging. The only existing results on quantum data either have a unique design specific to the 1-qubit case [13, 23], or suffer from robust training issues discussed below [4].

More importantly, classical GANs are well known for being delicate and somewhat unstable in training. In particular, it is known [1] that the choice of the metric between real and fake distributions will be critical for the stability of the performance in the training. A few widely used ones such as the Kullback-Leibler (KL) divergence, the Jensen-Shannon (JS) divergence, and the total variation (or statistical) distance are not sensible for learning distributions supported by low dimensional generative models. The shortcoming of these metrics will likely carry through to their quantum counterparts and hence quantum GANs based on these metrics will likely suffer from the same weaknesses in training. This training issue was not significant in the existing numerical study of quantum GANs in the 1-qubit case [13, 23]. However, as observed by [4] and us, the training issue becomes much more significant when the quantum system scales up, even just in the case of a few qubits.

To tackle the training issue of classical GANs, a lot of research has been conducted on the convergence of training GANs in classical machine learning. A seminal work [1] used *Wasserstein distance* (or, *optimal transport* distance) [43] as a metric for measuring the distance between real and fake distributions. Comparing to other measures (such as KL and JS), Wasserstein distance is more appealing from optimization perspective because of its continuity and smoothness. As a result, the corresponding Wasserstein GAN (WGAN) is promising for improving the training stability of GANs. There are a lot of subsequent studies on various modifications of the WGAN, such as GAN with regularized Wasserstein distance [35], WGAN with entropic regularizers [12, 38], WGAN with gradient penalty [20, 31], relaxed WGAN [21], etc. It is known [26] that WGAN and its variants such as [20] have demonstrated improved training stability compared to the original GAN formulation.

**Contributions.** Inspired by the success of classical Wasserstein GANs and the need of smooth, robust, and scalable training methods for quantum GANs on quantum data, we propose the first design of quantum Wasserstein GANs (qWGANs). To this end, our technical contributions are multi-folded.

In Section 3, we propose a quantum counterpart of the Wasserstein distance, denoted by $\mathrm{qW}(P, Q)$ between quantum data $P$ and $Q$, inspired by [1, 43]. We prove that $\mathrm{qW}(\cdot, \cdot)$ is a semi-metric (i.e., a metric without the triangle inequality) over quantum data and inherits nice properties such as continuity and smoothness of the classical Wasserstein distance. We will discuss a few other proposals of quantum Wasserstein distances such as [6, 8–10, 18, 29, 32, 45] and in particular why most of them are not suitable for the purpose of generating quantum data in GANs. We will also discuss the limitation of our proposal of quantum Wasserstein semi-metric and hope its successful application in quantum GANs could provide another perspective and motivation to study this topic.

In Section 4, we show how to add the quantum *entropic* regularization to $\mathrm{qW}(\cdot, \cdot)$ to further smoothen the loss function in the spirit of the classical case (e.g., [35]). We then show the construction of our regularized quantum Wasserstein GAN (qWGAN) in Figure 1 and describe the configuration and the parameterization of both the generator and the discriminator. Most importantly, we show that the evaluation of the loss function and the evaluation of the gradient of the loss function can be in principle efficiently implemented on quantum machines. This enables direct applications of classical training methods of GANs, such as alternating gradient-based optimization, to the quantum setting. It is a wide belief that classical computation cannot efficiently simulate quantum machines, in our case, the evaluation of the loss function and its gradient. Hence, the ability of evaluating them efficiently on quantum machines is *critical* for its scalability.

In Section 5, we supplement our theoretical results with experimental validations via classical simulation of qWGAN. Specifically, we demonstrate numerical performance of our qWGAN for quantum systems up to 8 qubits for pure states and up to 3 qubits for mixed states (i.e., mixture of pure states). Comparing to existing results [4, 13, 23], our numerical performance is more favorable in both the system size and its numerical stability. To give a rough sense, a single step in the classical

simulation of the 8-qubit system involves multiple multiplications of $2^8 \times 2^8$ matrices. Learning a mixed state is much harder than learning pure states (a reasonable classical analogue of their difference is the one between learning a Gaussian distribution and learning a mixture of Gaussian distributions [2]). We present the only result for learning a true mixed state up to 3 qubits.

Furthermore, following a specific 4-qubit generator that is recently implemented on an ion-trap quantum machine [48] and a reasonable noise model on the same machine [47], we simulate the performance of our qWGAN with noisy quantum operations. Our result suggests that qWGAN can tolerant a reasonable amount of noise in quantum systems and still converge. This shows the possibility of implementing qWGAN on near-term (NISQ) machines [33].

Finally, we demonstrate a real-world application of qWGAN to approximate useful quantum application with large circuits by small ones. qWGAN can be used to approximate a potentially complicated unknown quantum state by a simple one when using a reasonably simple generator. We leverage this property and the Choi-Jamiołkowski isomorphism [28] between quantum operations and quantum states to generate a simple state that approximates another Choi-Jamiołkowski state corresponding to potentially complicated circuits in real quantum applications. The closeness in two Choi-Jamiołkowski states of quantum circuits will translate to the average output closeness between two quantum circuits over random input states. Specifically, we show that the quantum Hamiltonian simulation circuit for 1-d 3-qubit Heisenberg model in [11] can be approximated by a circuit of 52 gates with an average output fidelity over 0.9999 and a worst-case error 0.15. The circuit based on the second order product formula will need ~11900 gates, however, with a worst-case error 0.001.

**Related results.** All existing quantum GANs [4, 13, 23, 25, 34, 39, 46, 49], no matter dealing with classical or quantum data, have not investigated the possibility of using the Wasserstein distance. The most relevant works to ours are [4, 13, 23] with specific GANs dealing with quantum data. As we discussed above, [13, 23] only discussed the 1-qubit case (both pure and mixed) and [4] discussed the pure state case (up to 6 qubits) but with the loss function being the quantum counterpart of the total variation distance. Moreover, the mixed state case in [13] is a labeled one: in addition to observing their mixture, one also gets a label of which pure state it is sampled from.

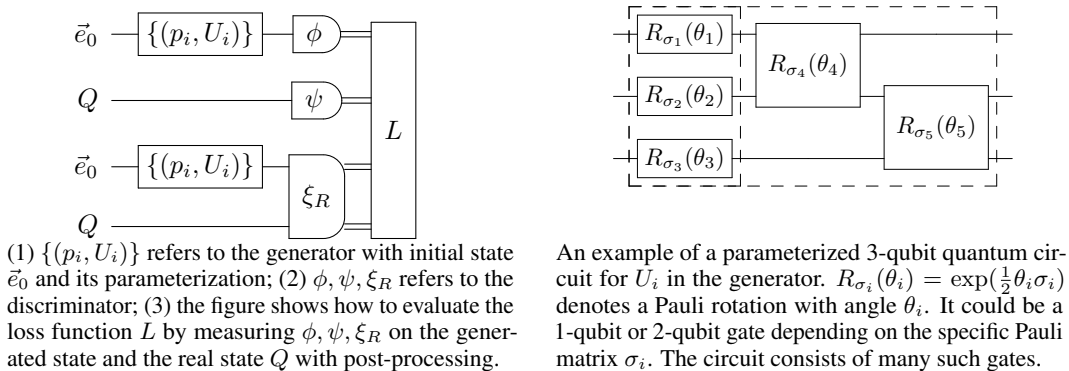

(1) $\{(p_i, U_i)\}$ refers to the generator with initial state $\vec{e}_0$ and its parameterization; (2) $\phi, \psi, \xi_R$ refers to the discriminator; (3) the figure shows how to evaluate the loss function $L$ by measuring $\phi, \psi, \xi_R$ on the generated state and the real state $Q$ with post-processing.

An example of a parameterized 3-qubit quantum circuit for $U_i$ in the generator. $R_{\sigma_i}(\theta_i) = \exp(\frac{1}{2}\theta_i\sigma_i)$ denotes a Pauli rotation with angle $\theta_i$. It could be a 1-qubit or 2-qubit gate depending on the specific Pauli matrix $\sigma_i$. The circuit consists of many such gates.

Figure 1: The Architecture of Quantum Wasserstein GAN.

## 2 Classical Wasserstein Distance & Wasserstein GANs

Let us first review the definition of Wasserstein distance and how it is used in classical WGANs.

**Wasserstein distance** Consider two probability distributions $p$ and $q$ given by corresponding density functions $p \colon \mathcal{X} \to \mathbb{R}, q \colon \mathcal{Y} \to \mathbb{R}$. Given a cost function $c \colon \mathcal{X} \times \mathcal{Y} \to \mathbb{R}$, the optimal transport cost between $p$ and $q$, known as the *Kantorovich's* formulation [43], is defined as

$$d_c(p,q) := \min_{\pi \in \Pi(p,q)} \int_{\mathcal{X}} \int_{\mathcal{Y}} \pi(x,y)c(x,y) \, \mathrm{d}x \, \mathrm{d}y \qquad (2.1)$$

where $\Pi(p,q)$ is the set of joint distributions $\pi$ having marginal distributions $p$ and $q$, i.e., $\int_{\mathcal{Y}} \pi(x,y) \, \mathrm{d}y = p(x)$ and $\int_{\mathcal{X}} \pi(x,y) \, \mathrm{d}x = q(y)$.

**Wasserstein GAN** The Wasserstein distance $d_c(p,q)$ can be used as an objective for learning a real distribution $q$ by a parameterized function $G_\theta$ that acts on a base distribution $p$. Then the objective

becomes learning parameters $\theta$ such that $d_c(G_\theta(p), q)$ is minimized as follows:

$$\min_\theta \min_{\pi \in \Pi(\mathcal{P}, \mathcal{Q})} \int_{\mathcal{X}} \int_{\mathcal{Y}} \pi(x, y) c(G_\theta(x), y) \, dx \, dy. \tag{2.2}$$

In [1], Arjovsky et al. propose using the dual of (2.2) to formulate the original min-min problem into a min-max problem, i.e., a generative adversarial network, with the following form:

$$\min_\theta \max_{\alpha, \beta} \quad \mathbb{E}_{x \sim \mathcal{P}}[\phi_\alpha(x)] - \mathbb{E}_{y \sim \mathcal{Q}}[\psi_\beta(y)], \tag{2.3}$$

$$\text{s.t} \quad \phi_\alpha(G_\theta(x)) - \psi_\beta(y) \leq c(G_\theta(x), y), \ \forall x, y, \tag{2.4}$$

where $\phi, \psi$ are functions parameterized by $\alpha, \beta$ respectively. This is advantageous because it is usually easier to parameterize functions rather than joint distributions. The constraint (2.4) is usually enforced by a regularizer term for actual implementation. Out of many choices of regularizers, the most relevant one to ours is the entropy regularizer in [35]. In the case that $c(x, y) = \|x - y\|_2$ and $\phi = \psi$ in (2.3), the constraint is that $\phi$ must be a 1-Lipschitz function. This is often enforced by the gradient penalty method in a neural network used to parameterize $\phi$.

## 3 Quantum Wasserstein Semimetric

**Mathematical formulation of quantum data** We refer curious readers to Supplemental Materials A for a more comprehensive introduction. Any quantum data (or quantum states) over space $\mathcal{X}$ (e.g., $\mathcal{X} = \mathbb{C}^d$) are mathematically described by a *density operator* $\rho$ that is a *positive semidefinite* matrix (i.e., $\rho \succeq 0$) with trace one (i.e., $\text{Tr}(\rho) = 1$), and the set of which is denoted by $\mathcal{D}(\mathcal{X})$.

A quantum state $\rho$ is *pure* if $\text{rank}(\rho) = 1$; otherwise it is a *mixed* state. For a pure state $\rho$, it can be represented by the outer-product of a *unit* vector $\vec{v} \in \mathbb{C}^d$, i.e., $\rho = \vec{v}\vec{v}^\dagger$, where $\dagger$ refers to conjugate transpose. We can also use $\vec{v}$ to directly represent pure states. Mixed states are a classical mixture of pure states, e.g., $\rho = \sum_i p_i \vec{v}_i \vec{v}_i^\dagger$ where $p_i$s form a classical distribution and $\vec{v}_i$s are all unit vectors.

Quantum states in a composed system of $\mathcal{X}$ and $\mathcal{Y}$ are represented by density operators $\rho$ over the Kronecker-product space $\mathcal{X} \otimes \mathcal{Y}$ with dimension $\dim(\mathcal{X}) \dim(\mathcal{Y})$. 1-qubit systems refer to $\mathcal{X} = \mathbb{C}^2$. A 2-qubit system has dimension 4 ($\mathcal{X}^{\otimes 2}$) and an $n$-qubit system has dimension $2^n$. The partial trace operation $\text{Tr}_\mathcal{X}(\cdot)$ (resp. $\text{Tr}_\mathcal{Y}(\cdot)$) is a linear mapping from $\rho$ to its marginal state on $\mathcal{Y}$ (resp. $\mathcal{X}$).

**From classical to quantum data** Classical distributions $p, q$ in (2.1) can be viewed as special mixed states $\mathcal{P} \in \mathcal{D}(\mathcal{X}), \mathcal{Q} \in \mathcal{D}(\mathcal{Y})$ where $\mathcal{P}, \mathcal{Q}$ are diagonal and $p, q$ (viewed as density vectors) take the diagonals of $\mathcal{P}, \mathcal{Q}$ respectively. Note that this is different from the conventional meaning of samples from classical distributions, which are random variables with the corresponding distributions.

This distinction is important to understand quantum data as the former (i.e., density operators) rather than the latter (i.e., samples) actually represents the entity of quantum data. This is because there are multiple ways (different quantum measurements) to read out classical samples out of quantum data for one fixed density operator. Mathematically, this is because density operators in general can have off-diagonal terms and quantum measurements can happen along arbitrary bases.

Consider $\mathcal{X}$ and $\mathcal{Y}$ from (2.1) being finite sets. We can express the classical Wasserstein distance (2.1) as a special case of the matrix formulation of quantum data. Precisely, we can replace the integral in (2.1) by summation, which can be then expressed by the trace of $\Pi C$ where $C$ is a diagonal matrix with $c(x, y)$ in the diagonal. $\pi$ is also a diagonal matrix expressing the coupling distribution $\pi(x, y)$ of $p, q$. Namely, $\pi$'s diagonal is $\pi(x, y)$ and satisfies the coupling marginal condition $\text{Tr}_\mathcal{Y}(\pi) = P$ and $\text{Tr}_\mathcal{X}(\pi) = Q$ where $P, Q$ are diagonal matrices with the distribution of $p, q$ in the diagonal, respectively. As a result, the Kantorovich's optimal transport in (2.1) can be reformulated as

$$d_c(p, q) := \min_\pi \text{Tr}(\pi C) \tag{3.1}$$

$$\text{s.t.} \quad \text{Tr}_\mathcal{Y}(\pi) = \text{diag}\{p(x)\}, \ \text{Tr}_\mathcal{X}(\pi) = \text{diag}\{q(y)\}, \ \pi \in \mathcal{D}(\mathcal{X} \otimes \mathcal{Y}),$$

where $C = \text{diag}\{c(x, y)\}$. Note that (3.1) is effectively a linear program.

**Quantum Wasserstein semimetric** Our matrix reformulation of the classical Wasserstein distance (2.1) suggests a naive extension to the quantum setting as follows. Let $\text{qW}(\mathcal{P}, \mathcal{Q})$ denote the quantum Wasserstein semimetric between $\mathcal{P} \in \mathcal{D}(\mathcal{X}), \mathcal{Q} \in \mathcal{D}(\mathcal{Y})$, which is defined by

$$\text{qW}(\mathcal{P}, \mathcal{Q}) := \min_\pi \text{Tr}(\pi C) \tag{3.2}$$

$$\text{s.t.} \quad \text{Tr}_\mathcal{Y}(\pi) = \mathcal{P}, \ \text{Tr}_\mathcal{X}(\pi) = \mathcal{Q}, \ \pi \in \mathcal{D}(\mathcal{X} \otimes \mathcal{Y}),$$

where $C$ is a matrix over $\mathcal{X} \otimes \mathcal{Y}$ that should refer to some cost-type function. The choice of $C$ is hence critical to make sense of the definition. First, matrix $C$ needs to be Hermitian (i.e., $C = C^\dagger$) to make sure that $\mathrm{qW}(\cdot, \cdot)$ is real. A natural attempt is to use $C = \mathrm{diag}\{c(x, y)\}$ from (3.1), which turns out to be significantly wrong. This is because $\mathrm{qW}(\vec{v}\vec{v}^\dagger, \vec{v}\vec{v}^\dagger)$ will be strictly greater than zero for random choice of unit vector $\vec{v}$ in that case. This demonstrates a crucial difference between classical and quantum data: *while classical information is always stored in the diagonal (or computational basis) of the space, quantum information can be stored off-diagonally (or in an arbitrary basis of the space)*. Thus, choosing a diagonal $C$ fails to detect the off-diagonal information in quantum data.

Our proposal is to leverage the concept of *symmetric subspace* in quantum information [22] to make sure that $\mathrm{qW}(P, P) = 0$ for any $P$. The projection onto the symmetric subspace is defined by

$$\Pi_{\mathrm{sym}} := \frac{1}{2}(\mathrm{I}_{\mathcal{X} \otimes \mathcal{Y}} + \mathrm{SWAP}), \qquad (3.3)$$

where $\mathrm{I}_{\mathcal{X} \otimes \mathcal{Y}}$ is the identity operator over $\mathcal{X} \otimes \mathcal{Y}$ and $\mathrm{SWAP}$ is the operator such that $\mathrm{SWAP}(\vec{x} \otimes \vec{y}) = (\vec{y} \otimes \vec{x}), \forall \vec{x} \in \mathcal{X}, \vec{y} \in \mathcal{Y}$.[2] It is well known that $\Pi_{\mathrm{sym}}(\vec{u} \otimes \vec{u}) = \vec{u} \otimes \vec{u}$ for all unit vectors $u$. With this property and by choosing $C$ to be the complement of $\Pi_{\mathrm{sym}}$, i.e.,

$$C := \mathrm{I}_{\mathcal{X} \otimes \mathcal{Y}} - \Pi_{\mathrm{sym}} = \frac{1}{2}(\mathrm{I}_{\mathcal{X} \otimes \mathcal{Y}} - \mathrm{SWAP}), \qquad (3.4)$$

we can show $\mathrm{qW}(P, P) = 0$ for any $P$. This is achieved by choosing $\pi = \sum_i \lambda_i(\vec{v}_i\vec{v}_i^\dagger \otimes \vec{v}_i\vec{v}_i^\dagger)$ given $P$'s spectral decomposition $P = \sum_i \lambda_i \vec{v}_i \vec{v}_i^\dagger$. Moreover, we can show

**Theorem 3.1** (Proof in Supplemental Materials B). $\mathrm{qW}(\cdot, \cdot)$ *forms a semimetric over $\mathcal{D}(\mathcal{X})$ over any space $\mathcal{X}$, i.e., for any $\mathcal{P}, \mathcal{Q} \in \mathcal{D}(\mathcal{X})$,*

1. *$\mathrm{qW}(\mathcal{P}, \mathcal{Q}) \geq 0$,*
2. *$\mathrm{qW}(\mathcal{P}, \mathcal{Q}) = \mathrm{qW}(\mathcal{Q}, \mathcal{P})$,*
3. *$\mathrm{qW}(\mathcal{P}, \mathcal{Q}) = 0$ iff $\mathcal{P} = \mathcal{Q}$.*

Even though our definition of $\mathrm{qW}(\cdot, \cdot)$, especially the choice of $C$, does not directly come from a cost function $c(x, y)$ over $\mathcal{X}$ and $\mathcal{Y}$, it however still encodes some geometry of the space of quantum states. For example, let $P = \vec{v}\vec{v}^\dagger$ and $Q = \vec{u}\vec{u}^\dagger$, $\mathrm{qW}(P, Q)$ becomes $0.5\,(1 - |\vec{u}^\dagger \vec{v}|^2)$ where $|\vec{u}^\dagger \vec{v}|$ depends on the angle between $\vec{u}$ and $\vec{v}$ which are unit vectors representing (pure) quantum states.

**The dual form of** $\mathrm{qW}(\cdot, \cdot)$  The formulation of $\mathrm{qW}(\cdot, \cdot)$ in (3.2) is given by a semidefinite program (SDP), opposed to the classical form in (3.1) given by a linear program. Its dual form is as follows.

$$\max_{\phi, \psi} \quad \mathrm{Tr}(Q\psi) - \mathrm{Tr}(P\phi) \qquad (3.5)$$

$$\mathrm{s.t.} \quad \mathrm{I}_{\mathcal{X}} \otimes \psi - \phi \otimes \mathrm{I}_{\mathcal{Y}} \preceq C, \phi \in \mathcal{H}(\mathcal{X}),\ \psi \in \mathcal{H}(\mathcal{Y}),$$

where $\mathcal{H}(\mathcal{X}), \mathcal{H}(\mathcal{Y})$ denote the set of Hermitian matrices over space $\mathcal{X}$ and $\mathcal{Y}$. We further show the *strong duality* for this SDP in Supplemental Materials B. Thus, both the primal (3.2) and the dual (3.5) can be used as the definition of $\mathrm{qW}(\cdot, \cdot)$.

**Comparison with other quantum Wasserstein metrics**  There have been a few different proposals that introduce matrices into the original definition of classical Wasserstein distance. We will compare these definitions with ours and discuss whether they are appropriate in our context of quantum GANs.

A few of these proposals (e.g., [7, 9, 10]) extend the dynamical formulation of Benamou and Brenier [3] in optimal transport to the matrix/quantum setting. In this formulation, couplings are defined not in terms of joint density measures, but in terms of smooth paths $t \to \rho(x, t)$ in the space of densities that satisfy some continuity equation with some time dependent vector field $v(x, t)$ inspired by physics. A pair $\{\rho(\cdot, \cdot), v(\cdot, \cdot)\}$ is said to couple $P$ and $Q$, the set of which is denoted $C(P, Q)$, if $\rho(x, t)$ is a smooth path with $\rho(\cdot, 0) = P$ and $\rho(\cdot, 1) = Q$. The 2-Wasserstein distance is

$$\mathrm{W}_2(P, Q) = \inf_{\{\rho(\cdot, \cdot), v(\cdot, \cdot)\} \in C(P, Q)} \frac{1}{2} \int_0^1 \int_{R^n} |v(x, t)|^2 \rho(x, t)\, \mathrm{d}x\, \mathrm{d}t. \qquad (3.6)$$

The above formulation seems difficult to manipulate in the context of GAN. It is unclear (a) whether the above definition has a favorable duality to admit the adversarial training and (b) whether physics-inspired quantities like $v(x, t)$ are suitable for the purpose of generating fake quantum data.

A few other proposals (e.g., [29, 32]) introduce the matrix-valued mass defined by a function $\mu :$ $X \to C^{n \times n}$ over domain $X$, where $\mu(x)$ is positive semidefinite and satisfies $\text{Tr}(\int_X \mu(x)dx) = 1$. Instead of considering transport probability masses from $X$ to $Y$, one considers transporting a matrix-valued mass $\mu_0(x)$ on $X$ to another matrix-valued mass $\mu_1(y)$ on $Y$. One can similarly define the Kantorovich's coupling $\pi(x, y)$ of $\mu_0(x)$ and $\mu_1(y)$, and define the Wasserstein distance based on a slight different combination of $\pi(x, y)$ and $c(x, y)$ comparing to (2.1). This definition, however, fails to derive a new metric between two matrices. This is because the defined Wasserstein distance still measures the distance between $X$ and $Y$ based on some induced measure ($\| \cdot \|_F$) on the dimension-$n$ matrix space. This is more evident when $X = \{P\}$ and $Y = \{Q\}$. The Wasserstein distance reduces to $c(x, y) + \|P - Q\|_F^2$ where the Frobenius norm ($\| \cdot \|_F$) is directly used in the definition.

The proposals in [6, 18] are very similar to us in the sense they define the same coupling in the Kantorovich's formulation as ours. However, their definition of the Wasserstein distance motivated by physics is induced by unbounded operator applied on continuous space, e.g., $\nabla_x, \text{div}_x$. This makes their definition only applicable to continuous space, rather than qubits in our setting.

The closest result to ours is [45], although the authors haven't proposed one concrete quantum Wasserstein metric. Instead, they formulate a general form of reasonable quantum Wasserstein metrics between finite-dimensional quantum states and prove that Kantorovich-Rubinstein theorem does not hold under this general form. Namely, they show the trace distance between quantum states cannot be determined by any quantum Wasserstein metric out of their general form.

**Limitation of our** $\text{qW}(\cdot, \cdot)$   Although we have successfully implemented qWGAN based on our $\text{qW}(\cdot, \cdot)$ and observed improved numerical performance, there are a few perspectives about $\text{qW}(\cdot, \cdot)$ worth further investigation. First, numerical study reveals that $\text{qW}(\cdot, \cdot)$ does not satisfy the triangle inequality. Second, our $\text{qW}(\cdot, \cdot)$ does not come from an explicit cost function, even though it encodes some geometry of the quantum state space. We conjecture that there could be a concrete underlying cost function and our $\text{qW}(\cdot, \cdot)$ (or a related form) could be emerged as the 2-Wasserstein metric of that cost function. We hope our work provides an important motivation to further study this topic.

## 4   Quantum Wasserstein GAN

We describe the specific architecture of our qWGAN (Figure 1) and its training. Similar to (2.2) with the fake state $P$ from a parameterized quantum generator $G$, consider

$$\min_G \min_\pi \quad \text{Tr}(\pi C) \tag{4.1}$$
$$\text{s.t.} \quad \text{Tr}_{\mathcal{Y}}(\pi) = P, \text{Tr}_{\mathcal{X}}(\pi) = Q, \pi \in \mathcal{D}(\mathcal{X} \otimes \mathcal{Y}),$$

or similar to (2.3) by taking the dual from (3.5),

$$\min_G \max_{\phi, \psi} \quad \text{Tr}(Q\psi) - \text{Tr}(P\phi) = \mathbb{E}_Q[\psi] - \mathbb{E}_P[\phi] \tag{4.2}$$
$$\text{s.t.} \quad \text{I}_{\mathcal{X}} \otimes \psi - \phi \otimes \text{I}_{\mathcal{Y}} \preceq C, \phi \in \mathcal{H}(\mathcal{X}), \ \psi \in \mathcal{H}(\mathcal{Y}),$$

where we abuse the notation of $\mathbb{E}_Q[\psi] := \text{Tr}(Q\psi)$, which refers to the expectation of the outcome of measuring Hermitian $\psi$ on quantum state $Q$. We hence refer $\phi, \psi$ as the discriminator.

**Regularized Quantum Wasserstein GAN**
The dual form (4.2) is inconvenient for optimizing directly due to the constraint $\text{I}_{\mathcal{X}} \otimes \psi - \phi \otimes \text{I}_{\mathcal{Y}} \preceq C$. Inspired by the entropy regularizer in the classical setting (e.g., [35]), we add a *quantum-relative-entropy-based* regularizer between $\pi$ and $P \otimes Q$ with a tunable parameter $\lambda$ to (4.1) to obtain

$$\min_G \min_\pi \quad \text{Tr}(\pi C) + \lambda \text{Tr}(\pi \log(\pi) - \pi \log(P \otimes Q)) \tag{4.3}$$
$$\text{s.t.} \quad \text{Tr}_{\mathcal{Y}}(\pi) = P, \text{Tr}_{\mathcal{X}}(\pi) = Q, \pi \in \mathcal{D}(\mathcal{X} \otimes \mathcal{Y}).$$

Using duality and the Golden-Thompson inequality [17, 40], we can approximate (4.3) by

$$\min_G \max_{\phi, \psi} \quad \mathbb{E}_Q[\psi] - \mathbb{E}_P[\phi] - \mathbb{E}_{P \otimes Q}[\xi_R] \quad \text{s.t.} \ \phi \in \mathcal{H}(\mathcal{X}), \ \psi \in \mathcal{H}(\mathcal{Y}), \tag{4.4}$$

where $\xi_R$ refers to the regularizing Hermitian

$$\xi_R = \frac{\lambda}{e} \exp\left(\frac{-C - \phi \otimes \text{I}_{\mathcal{Y}} + \text{I}_{\mathcal{X}} \otimes \psi}{\lambda}\right). \tag{4.5}$$

Similar to [35], we prove that this entropic regularization ensures that the objective for the outer minimization problem (4.4) is *differentiable* in $P$. (Proofs are given in Supplemental Materials B.2.)

**Parameterization of the Generator and the Discriminator**

**Generator** $G$ is a quantum operation that generates $P$ from a fixed initial state $\rho_0$ (e.g., the classical all-zero state $\vec{e}_0$). Specifically, generator $G$ can be described by an ensemble $\{(p_1, U_1), \ldots, (p_r, U_r)\}$ that means applying the unitary $U_i$ with probability $p_i$. The distribution $\{p_1, \ldots, p_r\}$ can be parameterized directly or through some classical generative network. The rank of the generated state is $r$ ($r = 1$ for pure states and $r > 1$ for mixed states). Our experiments include the cases $r = 1, 2$.

Each unitary $U_i$ refers to a quantum circuit consisting of simple parameterized 1-qubit and 2-qubit Pauli-rotation quantum gates (see the right of Figure 1). These Pauli gates can be implemented on near-term machines (e.g., [48]) and also form a universal gate set for quantum computation. Hence, this generator construction is widely used in existing quantum GANs. The $j$th gate in $U_i$ contains an angle $\theta_{i,j}$ as the parameter. All variables $p_i, \theta_{i,j}$ constitute the set of parameters for the generator.

**Discriminator** $\phi, \psi$ can be parameterized at least in two ways. The first approach is to represent $\phi, \psi$ as linear combinations of tensor products of Pauli matrices, which form a basis of the matrix space (details on Pauli matrices and measurements can be found in Supplemental Materials A). Let $\phi = \sum_k \alpha_k A_k$ and $\psi = \sum_l \beta_l B_l$, where $A_k, B_l$ are tensor products of Pauli matrices. To evaluate $\mathbb{E}_P[\phi]$ (similarly for $\mathbb{E}_Q[\psi]$), by linearity it suffices to collect the information of $\mathbb{E}_P[A_k]$s, which are simply Pauli measurements on the quantum state $P$ and amenable to experiments. Hence, $\alpha_k$ and $\beta_l$ can be used as the parameters of the discriminator. The second approach is to represent $\phi, \psi$ as parameterized quantum circuits (similar to the $G$) with a measurement in the computational basis. The set of parameters of $\phi$ (respectively $\psi$) could be the parameters of the circuit and values associated with each measurement outcome. Our implementation mostly uses the first representation.

**Training the Regularized Quantum Wasserstein GAN**

For the scalability of the training of the Regularized Quantum Wasserstein GAN, one must be able to evaluate the loss function $L = \mathbb{E}_Q[\psi] - \mathbb{E}_P[\phi] - \mathbb{E}_{P \otimes Q}[\xi_R]$ or its gradient efficiently on a quantum computer. Ideally, one would hope to directly approximate gradients by quantum computers to facilitate the training of qWGAN, e.g., by using the alternating gradient descent method. We show that it is indeed possible and we outline the key steps. More details are in Supplemental Materials C.

**Computing the loss function:** Each unitary operation $U_i$ that refers to an actual quantum circuit can be efficiently evaluated on quantum machines in terms of the circuit size. It can be shown that $L$ is a linear function of $P$ and can be computed by evaluating each $L_i = \mathbb{E}_Q[\psi] - \mathbb{E}_{U_i \rho_0 U_i^\dagger}[\phi] - \mathbb{E}_{U_i \rho_0 U_i^\dagger \otimes Q}[\xi_R]$ where $U_i \rho_0 U_i^\dagger$ refers to the state after applying $U_i$ on $\rho_0$. Similarly, one can show that $L$ is a linear function of the Hermitian matrices $\phi, \psi, \xi_R$. Our parameterization of $\phi$ and $\psi$ readily allows the use of efficient Pauli measurements to evaluate $\mathbb{E}_P[\phi]$ and $\mathbb{E}_Q[\psi]$. To handle the tricky part $\mathbb{E}_{P \otimes Q}[\xi_R]$, we relax $\xi_R$ and use a Taylor series to approximate $\mathbb{E}_{P \otimes Q}[\xi_R]$; the result form can again be evaluated by Pauli measurements composed with simple SWAP operations. As the major computation (e.g., circuit evaluation and Pauli measurements) is efficient on quantum machines, the overall implementation is efficient with possible overhead of sampling trials.

**Computing the gradients:** The parameters of the qWGAN are $\{p_i\} \cup \{\theta_{i,j}\} \cup \{\alpha_k\} \cup \{\beta_l\}$. $L$ is a linear function of $p_i, \alpha_k, \beta_l$. Thus it can be shown that the partial derivatives w.r.t. $p_i$ can be computed by evaluating the loss function on a generated state $U_i \rho_0 U_i^\dagger$ and the partial derivatives w.r.t. $\alpha_k, \beta_l$ can be computed by evaluating the loss function with $\phi, \psi$ replaced with $A_k, B_l$ respectively. The partial derivatives w.r.t. $\theta_{i,j}$ can be evaluated using techniques due to [36] via a simple yet elegant modification of the quantum circuits used to evaluate the loss function. The complexity analysis is similar to above. The only new ingredient is the quantum circuits to evaluate the partial derivatives w.r.t. $\theta_{i,j}$ due to [36], which are again efficient on quantum machines.

**Summary of the training complexity:** A rough complexity analysis above suggests that one step of the evaluation of the loss function (or the gradients) of our qWGAN can be efficiently implemented on quantum machines. (A careful analysis is in Supplemental Materials C.5.) Given this ability, the rest of the training of qWGAN is similar to the classical case and will share the same complexity. It is worthwhile mentioning that quantum circuit evaluation and Pauli measurements are not known to be efficiently computable by classical machines; the best known approach will cost exponential time.

# 5 Experimental Results

We supplement our theoretical findings with numerical results by classical simulation of quantum WGANs of learning *pure* states (up to 8 qubits) and *mixed* states (up to 3 qubits) as well as its performance on noisy quantum machines. We use quantum fidelity between the generated and target

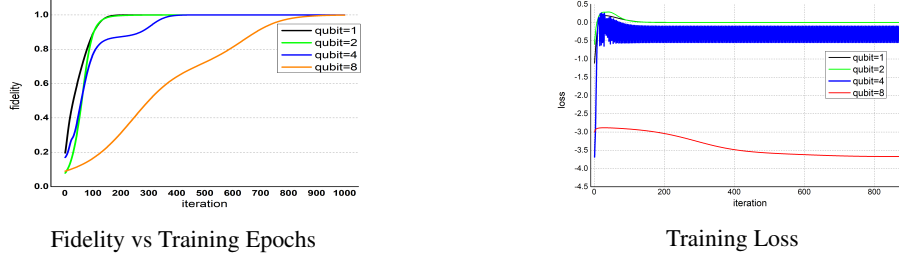

Fidelity vs Training Epochs          Training Loss

Figure 2: A typical performance of learning pure states (1,2,4, and 8 qubits).

states to track the progress of our quantum WGAN. If the training is successful, the fidelity will approach 1. Our quantum WGAN is trained using the alternating gradient descent method.

In most of the cases, the target state is generated by a circuit sharing the same structure with the generator but with randomly chosen parameters. We also demonstrate a special target state corresponding to useful quantum unitaries via Choi-Jamiołkowski isomorphism. More details of the following experiments (e.g., parameter choices) can be found in Supplemental Materials D.

Most of the simulations were run on a dual core Intel I5 processor with 8G memory. The 8-qubit pure state case was run on a Dual Intel Xeon E5-2697 v2 @ 2.70GHz processor with 128G memory. All source codes are publicly available at `https://github.com/yiminghwang/qWGAN`.

**Pure states** We demonstrate a typical performance of quantum WGAN of learning $1, 2, 4$, and $8$ qubit pure states in Figure 2. We also plot the average fidelity for 10 runs with random initializations in Figure 3 which shows the numerical stability of qWGAN.

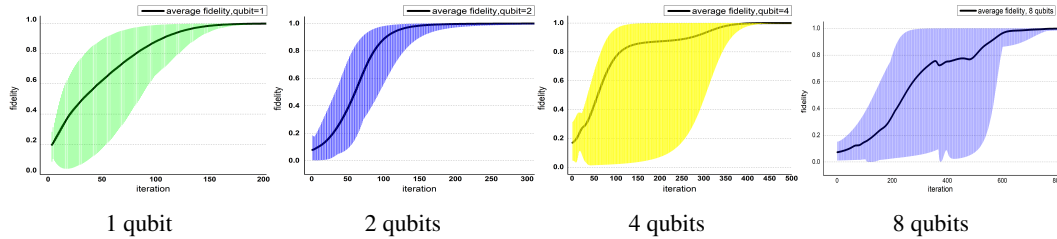

1 qubit                 2 qubits                 4 qubits                 8 qubits

Figure 3: Average performance of learning pure states (1, 2, 4, 8 qubits) where the black line is the average fidelity over multi-runs with random initializations and the shaded area refers to the range of the fidelity.

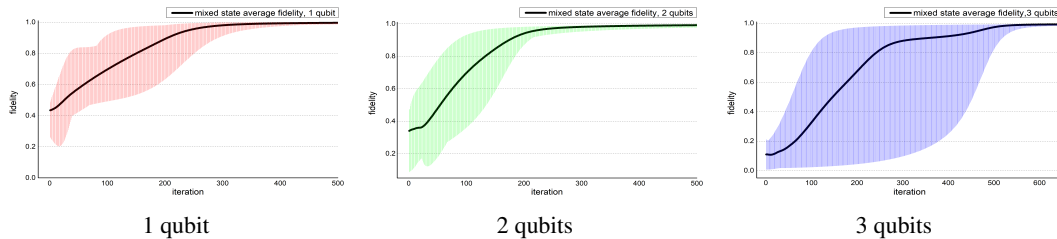

1 qubit                      2 qubits                      3 qubits

Figure 4: Average performance of learning mixed states (1, 2, 3 qubits) where the black line is the average fidelity over multi-runs with random initializations and the shaded area refers to the range of the fidelity.

**Mixed states** We also demonstrate a typical learning of mixed quantum states of rank 2 with $1, 2$, and $3$ qubits in Figure 4. The generator now consists of 2 unitary operators and 2 real probability parameters $p_1, p_2$ which are normalized to form a probability distribution using a softmax layer.

**Learning pure states with noise** To investigate the possibility of implementing our quantum WGAN on near-term machines, we perform a numerical test on a practically implementable 4-qubit generator on the ion-trap machine [48] with an approximate noise model [47]. We deem this as the closest example that we can simulate to an actual physical experiment. In particular, we add a Gaussian sampling noise with standard deviation $\sigma = 0.2, 0.15, 0.1, 0.05$ to the measurement outcome of the quantum system. Our results (in Figure 5) show that the quantum WGAN can still

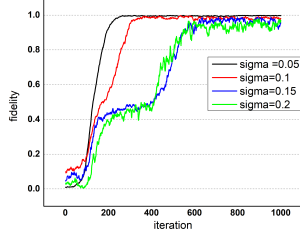

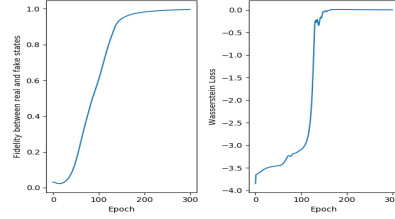

Figure 5: Learning 4-qubit pure states with noisy quantum operations.

Figure 6: Learning to approximate the 3-qubit Hamiltonian simulation circuit of the 1-d Heisenberg model.

learn a 4-qubit pure state in the presence of this kind of noise. As expected, noise of higher degrees (higher $\sigma$) increases the number of epochs before the state is learned successfully.

**Comparison with existing experimental results**   We will compare to quantum GANs with quantum data [4, 13, 23]. It is unfortunate that there is neither precise figure nor public data in their papers which makes a precise comparison infeasible. However, we manage to give a rough comparison as follows. Ref. [13] studies the pure state and the labeled mixed state case for 1 qubit. It can be inferred from the plots of their results (Figure 8.b in [13]) that the relative entropy for both labels converges to $10^{-10}$ after $\sim 5000$ iterations, and it takes more than 1000 iterations for the relative entropy to significantly decrease from 1. Ref. [23] performs experiments to learn 1-qubit pure and mixed states using a quantum GAN on a superconducting quantum circuit. However, the specific design of their GAN is very unique to the 1-qubit case. They observe that the fidelity between the fake state and the real state approaches 1 after 220 iterations for the pure state, and 120 iterations for the mixed state. From our figures, qWGAN can quickly converge for 1-qubit pure states after $150-160$ iterations and for a 1-qubit mixed state after $\sim 120$ iterations.

Ref. [4] studies only pure states but with numerical results up to 6 qubits. In particular, they demonstrate (in Figure 6 from [4]) in the case of 6-qubit that the normal gradient descent approach, like the one we use here, won't make much progress at all after 600 iterations. Hence they introduce a new training method. This is in sharp contrast to our Figure 2 where we demonstrate smooth convergence to fidelity 1 with the simple gradient descent for 8-qubit pure states within 900 iterations.

**Application: approximating quantum circuits**   To approximate any quantum circuit $U_0$ over $n$-qubit space $\mathcal{X}$, consider Choi-Jamiołkowski state $\Psi_0$ over $\mathcal{X} \otimes \mathcal{X}$ defined as $(U_0 \otimes \mathrm{I}_{\mathcal{X}})\Phi$ where $\Phi$ is the maximally entangled state $\frac{1}{\sqrt{2^n}} \sum_{i=0}^{2^n-1} \vec{e_i} \otimes \vec{e_i}$ and $\{e_i\}_{i=0}^{2^n-1}$ forms an orthonormal basis of $\mathcal{X}$. The generator is the normal generator circuit $U_1$ on the first $\mathcal{X}$ and identity on the second $\mathcal{X}$, i.e., $U_1 \otimes \mathrm{I}$. In order to learn for the 1-d 3-qubit Heisenberg model circuit (treated as $U_0$) in [11], we simply run our qWGAN to learn the 6-qubit Choi-Jamiołkowski state $\Psi_0$ in Figure 6 and obtain the generator (i.e., $U_1$). We use the gate set of single or 2-qubit Pauli rotation gates. Then $U_1$ only has 52 gates, while using the best product-formula (2nd order) $U_0$ has $\sim 11900$ gates. It is worth noting that $U_1$ achieves an average output fidelity over 0.9999 and a worst-case error 0.15, whereas $U_0$ has a worst-case error 0.001. However, the worst-case input of $U_1$ is not realistic in current experiments and hence the high average fidelity implies very reasonable approximation in practice.

## 6   Conclusion & Open Questions

We provide the first design of quantum Wasserstein GANs, its performance analysis on realistic quantum hardware through classical simulation, and a real-world application in this paper. At the technical level, we propose a counterpart of Wasserstein metric between quantum data. We believe that our result opens the possibility of quite a few future directions, for example:

- Can we implement our quantum WGAN on an actual quantum computer? Our noisy simulation suggests the possibility at least on an ion-trap machine.
- Can we apply our quantum WGAN to even larger and noisy quantum systems? In particular, can we approximate more useful quantum circuits using small ones by using quantum WGAN? It seems very likely but requires more careful numerical analysis.
- Can we better understand and build a rich theory of quantum Wasserstein metrics in light of [43]?

## Acknowledgement

We thank anonymous reviewers for many constructive comments and Yuan Su for helpful discussions about the reference [11]. SC, TL, and XW received support from the U.S. Department of Energy, Office of Science, Office of Advanced Scientific Computing Research, Quantum Algorithms Teams program. SF received support from Capital One and NSF CDS&E-1854532. TL also received support from an IBM Ph.D. Fellowship and an NSF QISE-NET Triplet Award (DMR-1747426). XW also received support from NSF CCF-1755800 and CCF-1816695.

## Footnotes

[2]One needs that $\mathcal{X}$ is isometric to $\mathcal{Y}$ to well define $\Pi_{\mathrm{sym}}$. However, this is without loss of generality by choosing appropriate and potentially larger spaces $\mathcal{X}$ and $\mathcal{Y}$ to describe quantum data.

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
