[Supplementary Material · QWGAN-supplementary.pdf]

# Supplementary Materials

## A  Preliminaries

### A.1  Quantum Information

We introduce necessary quantum information backgrounds for our qWGAN.

**Quantum states**  Quantum information can be formulated in terms of linear algebra. Given the space $\mathbb{C}^d$, its computational basis is denoted as $\{\vec{e}_0, \ldots, \vec{e}_{d-1}\}$, where $\vec{e}_i = (0, \ldots, 1, \ldots, 0)^\dagger$ with the $(i+1)^{\text{th}}$ entry being 1 and other entries being 0; here '$\dagger$' denotes the complex conjugate of a vector/matrix.

*Pure quantum states* with dimension $d$ are represented by unit vectors in $\mathbb{C}^d$: i.e., a vector $\vec{v} = (v_0, \ldots, v_{d-1})^\dagger$ is a quantum state if $\sum_{i=0}^{d-1} |v_i|^2 = 1$. For each $i$, $v_i$ is called the *amplitude* in $\vec{e}_i$. If there are at least two non-zero amplitudes, quantum state $\vec{v}$ is in *superposition* of the computational basis, a fundamental feature in quantum mechanics.

*Mixed quantum states* are probabilistic mixtures of pure quantum states. Formally, a mixed state can be written as $\sum_{k=1}^r p_k \vec{v}_k \vec{v}_k^\dagger$ where $p_k \geq 0 \ \forall k \in [r]$, $\sum_{k=1}^r p_k = 1$, and $\vec{v}_k$ is a pure state (i.e. $\|\vec{v}_k\|_2 = 1$) for all $k \in [r]$. Denote $\rho := \sum_{k=1}^r p_k \vec{v}_k \vec{v}_k^\dagger$; $\rho$ satisfies $\rho \succeq 0$, $\text{Tr}[\rho] = 1$, and $\rho^\dagger = \rho$ (i.e., $\rho$ is a *Hermitian matrix*). Such matrices are called *density matrices*, and every mixed state is a density matrix (and vice versa).

In many scenarios, quantum states are naturally composed of two parts. This comes to the concept of *bipartite quantum systems*, where a bipartite quantum state $\rho_{12}$ in $\mathbb{C}^{d_1} \otimes \mathbb{C}^{d_2}$ ($d_1, d_2 \in \mathbb{N}$) can be written as $\rho_{12} = \sum_i c_i \rho_{i,1} \otimes \rho_{i,2}$ for a probability distribution $\{c_i\}$ and density matrices $\{\rho_{i,1}\}$ in $\mathbb{C}^{d_1}$ and $\{\rho_{i,2}\}$ in $\mathbb{C}^{d_2}$. Since $\sum_i c_i = 1$ we have $\text{Tr}[\rho_{12}] = 1$, i.e., $\rho_{12}$ is a density matrix in $\mathbb{C}^{d_1} \otimes \mathbb{C}^{d_2}$; *partial trace* is defined to further characterize the properties in each separate part. Formally, the partial trace on system 1 is defined as $\text{Tr}_1[\rho_{12}] := \sum_i c_i \rho_{i,2}$, whereas the partial trace on system 2 is defined as $\text{Tr}_2[\rho_{12}] := \sum_i c_i \rho_{i,1}$.

**Qubits**  The basic element in classical computers is one bit, whereas the basic element in quantum computers is one *qubit*. Mathematically, a 1-qubit state is a state in $\mathbb{C}^2$ and can be written as $a\vec{e}_0 + b\vec{e}_1$ for some $a, b \in \mathbb{C}$ such that $|a|^2 + |b|^2 = 1$. An $n$-qubit state can be written as $\vec{v}_1 \otimes \cdots \otimes \vec{v}_n$ where each $\vec{v}_i$ ($i \in [n]$) is a qubit state, and $\otimes$ is the Kronecker product: if $\vec{u} \in \mathbb{C}^{d_1}$ and $\vec{v} \in \mathbb{C}^{d_2}$, then $\vec{u} \otimes \vec{v} \in \mathbb{C}^{d_1} \otimes \mathbb{C}^{d_2}$ is

$$\vec{u} \otimes \vec{v} = (u_0 v_0, u_0 v_1, \ldots, u_{d_1-1} v_{d_2-1})^\dagger. \tag{A.1}$$

$n$-qubit states are in a Hilbert space of dimension $2^n$.

**Unitary gates**  Having the definition of quantum states, it comes to the rules of their evolution. Note that we want to keep the quantum states normalized under $\ell_2$-norm; in linear algebra such transformations are known as *unitary transformation*. Formally, a matrix $U$ is unitary iff $UU^\dagger = I$.

The gates in quantum computation are always unitary gates and can be stated in the circuit model[3] where an *n-qubit gate* is a unitary matrix in $\mathbb{C}^{2^n}$. A common group of unitary gates on a qubit is the *Pauli gates*, where

$$\sigma_I = \begin{bmatrix} 1 & 0 \\ 0 & 1 \end{bmatrix}, \ \sigma_x = \begin{bmatrix} 0 & 1 \\ 1 & 0 \end{bmatrix}, \ \sigma_y = \begin{bmatrix} 0 & -i \\ i & 0 \end{bmatrix}, \ \sigma_z = \begin{bmatrix} 1 & 0 \\ 0 & -1 \end{bmatrix}; \tag{A.2}$$

note that the Pauli gates form a basis of all the unitaries acting on $\mathbb{C}^2$. Furthermore, $\sigma_I^2 = \sigma_x^2 = \sigma_y^2 = \sigma_z^2 = I$; this implies that the exponentiation of a Pauli matrix is a linear combination of Pauli matrices: for any phase $\theta \in \mathbb{R}$ and $\sigma \in \{\sigma_I, \sigma_x, \sigma_y, \sigma_z\}$, the Taylor expansion of $e^{\theta \sigma}$ is

$$e^{\theta \sigma} = \sum_{k=0}^{\infty} \frac{\theta^k \sigma^k}{k!} = \sum_{k=0}^{\infty} \frac{\theta^{2k}}{(2k)!} I + \sum_{k=0}^{\infty} \frac{\theta^{2k+1}}{(2k+1)!} \sigma. \tag{A.3}$$

**Quantum measurements**  Quantum states can be measured by quantum measurements. For pure states, the simplest measurement is to measure in the computational basis; for $\vec{v} = (v_1, \ldots, v_n)$, such measurement returns $k$ with probability $|v_k|^2$ for all $k \in [n]$. Recall that $\vec{v}$ is normalized such that $\|\vec{v}\|_2 = 1$, the measurement outcome constitutes a probability distribution on $[n]$. For $n$-qubit pure states $\vec{v}$, a common measurement is the *Pauli measurement*, where you first apply $\vec{v}$ by a tensor of Pauli gates $\sigma_1 \otimes \cdots \otimes \sigma_n$ ($\sigma_1, \ldots, \sigma_n \in \{\sigma_I, \sigma_x, \sigma_y, \sigma_z\}$) and measure in the computational basis $\{\vec{e_0} \otimes \cdots \otimes \vec{e_0}, \ldots, \vec{e_1} \otimes \cdots \otimes \vec{e_1}\}$.

For a density matrix $\rho$, the most general measurements are positive-operator valued measurements (POVMs), characterized by a set of Hermitian operators $\{E_1, \ldots, E_k\}$ such that 1) $E_i \succeq 0$ for all $i \in [k]$, and 2) $\sum_{i=1}^{k} E_i = I$. The outcome of the measurement is $i$ with probability $\mathrm{Tr}[\rho E_i]$; this also constitutes a probability distribution as $\sum_{i=1}^{k} \mathrm{Tr}[\rho E_i] = \mathrm{Tr}[\rho] = 1$.

**Distance measure**  There are various of ways to define the distance between two quantum states $\rho_1$ and $\rho_2$. One natural distance is the *trace distance* defined by $F_{\mathrm{Tr}}(\rho_1, \rho_2) := \mathrm{Tr}\,|\rho_1 - \rho_2|$, the sum of the absolute value of the eigenvalues of $\rho_1 - \rho_2$; this generalizes the *total variation distance* between classical distributions. Another common distance is the *fidelity*: $F(\rho_1, \rho_2) := \mathrm{Tr}[\sqrt{\sqrt{\rho_1}\rho_2\sqrt{\rho_1}}]^2$. $F(\rho, \sigma) = 1$ if and only if $\rho = \sigma$, and $F(\rho, \sigma)$ approaches 1 as $\rho$ approaches $\sigma$ [28].

Besides symmetric distances, people also consider divergences as they also characterize natural properties between two distributions. One such example is the *Kullback-Leibler divergence* (KL divergence) [24], also known as the *relative entropy*, defined as follows for two classical distributions $p$ and $q$ on $[n]$:

$$D_{\mathrm{KL}}(p\|q) = \sum_{i=1}^{n} p_i \log(p_i/q_i) = \sum_{i=1}^{n} p_i \log p_i - \sum_{i=1}^{n} p_i \log q_i. \tag{A.4}$$

Quantumly there is a natural extension, namely the *quantum relative entropy*, defined as follows:

$$S(\rho\|\sigma) := \mathrm{Tr}[\rho(\log \rho - \log \sigma)]. \tag{A.5}$$

(See (A.9) below for the definition of $\log \rho$ and $\log \sigma$.)

To learn quantum distributions (states), one must minimize some measure of *distance* between the true density matrix and our learned state; however, it turns out that the trace distance and the fidelity are not easily amenable to be optimized. This is the main reason why we adopt our quantum Wasserstein semimetric; see more discussions in Section 3 and Supplemental Materials B.

**Symmetric subspace**  Recall that our quantum Wasserstein semimetric in Section 3 is *symmetric*; achieving this requires the theory of *symmetric subspaces*. Given two Hilbert spaces $\mathcal{X}$ and $\mathcal{Y}$ that are isometric, a symmetric subspace of the space $\mathcal{X} \otimes \mathcal{Y}$ is the space of those vectors that are invariant to a permutation of $\mathcal{X}$ and $\mathcal{Y}$ individually. Ref. [22] proved that the projection onto the symmetric subspace is given by

$$\Pi_{\mathrm{sym}} := \frac{I + \mathrm{SWAP}}{2} \tag{A.6}$$

where I is the identity operator and SWAP is the operator such that $\mathrm{SWAP}(x \otimes y) = (y \otimes x), \forall x \in \mathcal{X}, y \in \mathcal{Y}$. It is also well known that $\Pi_{\mathrm{sym}}$ is a projector on $\mathcal{X} \otimes \mathcal{Y}$, ie. $\Pi_{\mathrm{sym}}^2 = \Pi_{\mathrm{sym}}$, and that $\Pi_{\mathrm{sym}}(u \otimes u) = u \otimes u$ for all quantum states $u$. This motivates us to choose the cost matrix $C$ in (4.1) to be the complement of the symmetric subspace, i.e.,

$$C := \frac{I - \mathrm{SWAP}}{2}. \tag{A.7}$$

Such choice is natural because on the one hand it ensures that $\mathrm{qW}(\rho, \rho) = 0$ for any quantum state $\rho$, and on the other hand it promises the symmetry of the semimetric, i.e., $\mathrm{qW}(\rho, \sigma) = \mathrm{qW}(\sigma, \rho)$ for any quantum states $\rho, \sigma$.

## A.2  Matrix Arithmetics

Unless otherwise mentioned, the matrices we consider are *Hermitian*, defined as all matrices $A$ such that $A^\dagger = A$. For any two Hermitian matrices $A, B \in \mathbb{C}^{n \times n}$, we say $A \succeq B$ iff $A - B$ is a positive semidefinite matrix (i.e., $A - B$ only has nonnegative eigenvalues), and $A \succ B$ iff $A - B$ is a positive definite matrix (i.e., $A - B$ only has positive eigenvalues).

A function of a Hermitian matrix is computed by taking summations of matrix powers under its Taylor expansion; for instance, for any Hermitian $A$ we have

$$\exp(A) := \sum_{k=0}^{\infty} \frac{A^k}{k!}, \tag{A.8}$$

and for any $0 \prec B \prec 2I$ we have

$$\log(B) := \sum_{k=1}^{\infty} \frac{(-1)^{k+1}}{k} (B - I)^k. \tag{A.9}$$

Furthermore, we introduce two tools for matrix arithmetics that we frequently use throughout the paper. The first is a rule for taking gradients of matrix functions:

**Lemma A.1** ([42]). *Given a Hermitian matrix $W \in \mathbb{C}^{n \times n}$ and a function $f \colon \mathbb{R} \to \mathbb{R}$, we define the gradient $\nabla_W f(W)$ as the entry-wise derivatives, i.e., $\nabla_W f(W) := (\frac{\partial f(W)_{ij}}{\partial W_{ij}})_{i,j=1}^n$. Then we have*

$$\nabla_W \operatorname{Tr}(W \log(W)) = [\log(W) + (W)]^\dagger = \log(W) + W. \tag{A.10}$$

For exponentiations of Hermitian matrices, we use the Golden-Thompson inequality stated as follows:

**Lemma A.2** ([17, 40]). *For any Hermitian matrices $A, B \in \mathbb{C}^{n \times n}$,*

$$\operatorname{Tr}(\exp(A + B)) \leq \operatorname{Tr}(\exp(A) \exp(B)). \tag{A.11}$$

# B    Properties of the Quantum Wasserstein Semimetric

## B.1    Proofs

**Lemma B.1.** *Strong Duality holds for the semidefinite program (3.2).*

*Proof.* Note that $\pi = \mathcal{P} \otimes \mathcal{Q}$ is a feasible solution to the primal program (3.2).

Consider the solution $\psi = -I_\mathcal{Y}$, $\phi = I_\mathcal{X}$ for the dual program (3.2). Then $I_\mathcal{X} \otimes \psi - \phi \otimes I_\mathcal{Y} - C = -2I_\mathcal{X} \otimes I_\mathcal{Y} - C$. For any vector $v \in \mathcal{X} \otimes \mathcal{Y}$, $v^\dagger(-2I_\mathcal{X} \otimes I_\mathcal{Y} - C)v = -2 - v^\dagger C v \leq -2 < 0$. Therefore $I_\mathcal{X} \otimes \psi - \phi \otimes I_\mathcal{Y} \prec C$ and the solution is strictly feasible. Since a strictly feasible solution exists to the dual program and the primal feasible set is non-empty, Slater's conditions are satisfied and the lemma holds [44, Theorem 1 (1)]. $\qquad\square$

Lemma B.1 shows that the primal and dual SDPs have the same optimal value and thus (3.5) can be taken as an alternate definition of the Quantum Wasserstein distance.

The following theorem establishes some properties of the Quantum Wasserstein distance.

**Theorem B.1.** $\mathrm{qW}(\cdot, \cdot)$ *forms a semimetric over the set of density matrices $\mathcal{D}(\mathcal{X})$ over any space $\mathcal{X}$, i.e., for any $\mathcal{P}, \mathcal{Q} \in \mathcal{D}(\mathcal{X})$,*

1. $\mathrm{qW}(\mathcal{P}, \mathcal{Q}) \geq 0$,
2. $\mathrm{qW}(\mathcal{P}, \mathcal{Q}) = \mathrm{qW}(\mathcal{Q}, \mathcal{P})$,
3. $\mathrm{qW}(\mathcal{P}, \mathcal{Q}) = 0$ *iff $\mathcal{P} = \mathcal{Q}$.*

*Proof.* We will use the definition of $\mathrm{qW}(\cdot, \cdot)$ from (3.2) with $\mathcal{Y}$ being an isometric copy of $\mathcal{X}$.

1. Consider the matrix $C = \frac{\mathrm{I} - \mathrm{SWAP}}{2}$. Let $\vec{u} = \sum_{i,j \in \Gamma} u_{ij} \vec{e}_i \vec{e}_j$ be any vector in $\mathcal{X} \otimes \mathcal{Y} = \mathbb{C}^{|\Gamma|} \otimes \mathbb{C}^{|\Gamma|}$. By simple calculation,

$$\vec{u}^\dagger C \vec{u} = \sum_{i,j} u_{ij}^*(u_{ij} - u_{ji}) = \sum_{i \leq j} (u_{ij}^* - u_{ji}^*)(u_{ij} - u_{ji}) = \sum_{i \leq j} |u_{ij} - u_{ji}|^2 \geq 0; \tag{B.1}$$

   thus $C$ is positive semidefinite. As a result, $\operatorname{Tr}(\pi C) \geq 0$ for all $\pi \succeq 0$, and $\mathrm{qW}(\mathcal{P}, \mathcal{Q}) \geq 0$ for all density matrices $\mathcal{P}, \mathcal{Q} \in \mathcal{D}(\mathcal{X})$.

2. This property trivially holds because of the definition in (3.2) is symmetric in $\mathcal{P}$ and $\mathcal{Q}$.

3. Suppose that $P = Q$ have spectral decomposition $\sum_i \lambda_i \vec{v}_i \vec{v}_i^\dagger$. Consider $\pi_0 = \sum_i \lambda_i (\vec{v}_i \vec{v}_i^\dagger \otimes \vec{v}_i \vec{v}_i^\dagger)$. Then, $\operatorname{Tr}(\pi_0 C) = \operatorname{Tr}(\sum_i \lambda_i (\vec{v}_i \vec{v}_i^\dagger \otimes \vec{v}_i \vec{v}_i^\dagger) C) = \operatorname{Tr}(\sum_i \lambda_i (\vec{v}_i^\dagger \otimes \vec{v}_i^\dagger) C(\vec{v}_i \otimes \vec{v}_i))$. Since $C = \frac{\mathrm{I} - \mathrm{SWAP}}{2}$, $C(\vec{v}_i \otimes \vec{v}_i) = 0$. Thus $\operatorname{Tr}(\pi_0 C) = 0$ and since $C$ is positive semidefinite, this must be the minimum. Thus $\mathrm{qW}(\mathcal{P}, \mathcal{P}) = 0$. $\qquad\square$

## B.2 Regularized Quantum Wasserstein Distance

The regularized primal version of the Quantum Wasserstein GAN is constructed from (4.1) by adding the relative entropy between the optimization variable $\pi$ and the joint distribution of the real and fake states $P \otimes Q$, given by $S(\pi \| P \otimes Q) = \mathrm{Tr}(\pi \log(\pi) - \pi \log(P \otimes Q))$:

$$\min_{\pi} \quad \mathrm{Tr}(\pi C) + \lambda \mathrm{Tr}(\pi \log(\pi) - \pi \log(P \otimes Q)) \tag{B.2}$$

$$\text{s.t.} \quad \mathrm{Tr}_{\mathcal{Y}}(\pi) = P, \mathrm{Tr}_{\mathcal{X}}(\pi) = Q, \pi \in \mathcal{D}(\mathcal{X} \otimes \mathcal{Y}).$$

Here $\lambda$ is a parameter that is chosen during training, and determines the weight given to the regularizer. To formulate the dual, we use Hermitian Lagrange multipliers $\phi$ and $\psi$ to construct a saddle point problem:

$$\min_{\pi} \max_{\psi,\phi} \quad \mathrm{Tr}(\pi C) + \lambda \mathrm{Tr}(\pi \log(\pi) - \pi \log(P \otimes Q))$$

$$+ \mathrm{Tr}(\phi(\mathrm{Tr}_{\mathcal{Y}}(\pi) - P)) - \mathrm{Tr}(\psi(\mathrm{Tr}_{\mathcal{X}}(\pi) - Q))$$

$$= \min_{\pi} \max_{\psi,\phi} \quad \mathrm{Tr}(\pi(C + \phi \otimes \mathrm{I}_{\mathcal{Y}} - \mathrm{I}_{\mathcal{X}} \otimes \psi)) - \mathrm{Tr}(P\phi)$$

$$+ \mathrm{Tr}(Q\psi) + \lambda \mathrm{Tr}(\pi \log(\pi) - \pi \log(P \otimes Q)). \tag{B.3}$$

Switching the order of the optimizations:

$$\max_{\psi,\phi} \min_{\pi} \quad \mathrm{Tr}(\pi(C + \phi \otimes \mathrm{I}_{\mathcal{Y}} - \mathrm{I}_{\mathcal{X}} \otimes \psi)) - \mathrm{Tr}(P\phi)$$

$$+ \mathrm{Tr}(Q\psi) + \lambda \mathrm{Tr}(\pi \log(\pi) - \pi \log(P \otimes Q)). \tag{B.4}$$

Solving the inner optimization problem for $\pi$ and using Lemma A.1, we have that for the optimal $\pi$,

$$(C + \phi \otimes \mathrm{I}_{\mathcal{Y}} - \mathrm{I}_{\mathcal{X}} \otimes \psi) + \lambda \log(\pi) + \lambda \mathrm{I} - \log(P \otimes Q) = 0. \tag{B.5}$$

Thus the dual optimization problem reduces to

$$\max_{\phi,\psi} \quad \mathrm{Tr}(Q\psi) - \mathrm{Tr}(P\phi) - \frac{\lambda}{e} \mathrm{Tr}\left(\exp\left(\frac{\log(P \otimes Q) - C - \phi \otimes \mathrm{I}_{\mathcal{Y}} + \mathrm{I}_{\mathcal{X}} \otimes \psi}{\lambda}\right)\right) \tag{B.6}$$

$$\text{s.t.} \quad \phi \in \mathcal{H}(\mathcal{X}), \psi \in \mathcal{H}(\mathcal{Y}).$$

Note that the additional term in the objective of the dual cannot be directly written as the expected value of measuring a Hermitian operator. However, we can use the Golden-Thompson inequality (Lemma A.2) to upper bound on the objective, which can be written in terms of the expectation as

$$\max_{\phi,\psi} \quad \mathrm{Tr}(Q\psi) - \mathrm{Tr}(P\phi) - \frac{\lambda}{e} \mathrm{Tr}\left((P \otimes Q)\exp\left(\frac{-C - \phi \otimes \mathrm{I}_{\mathcal{Y}} + \mathrm{I}_{\mathcal{X}} \otimes \psi}{\lambda}\right)\right)$$

$$= \max_{\phi,\psi} \mathbb{E}_Q[\psi] - \mathbb{E}_P[\phi] - \frac{\lambda}{e} \cdot \mathbb{E}_{P \otimes Q}\left[\exp\left(\frac{-C - \phi \otimes \mathrm{I}_{\mathcal{Y}} + \mathrm{I}_{\mathcal{X}} \otimes \psi}{\lambda}\right)\right] \tag{B.7}$$

$$\text{s.t.} \quad \phi \in \mathcal{H}(\mathcal{X}), \ \psi \in \mathcal{H}(\mathcal{Y}).$$

The regularized optimization problem has the following property:

**Lemma B.2.** *Let* $f \colon \mathcal{D}(\mathcal{X}) \to \mathbb{R}$ *be defined as*

$$\mathbb{E}_Q[\psi] - \mathbb{E}_P[\phi] - \frac{\lambda}{e} \cdot \mathbb{E}_{P \otimes Q}\left[\exp\left(\frac{-C - \phi \otimes \mathrm{I}_{\mathcal{Y}} + \mathrm{I}_{\mathcal{X}} \otimes \psi}{\lambda}\right)\right] \tag{B.8}$$

$$\text{s.t.} \quad \phi \in \mathcal{H}(\mathcal{X}), \ \psi \in \mathcal{H}(\mathcal{Y}).$$

*Then* $f(P)$ *is a differentiable function of* $P$.

*Proof.* The optimization objective (B.8) is clearly convex with respect to its parameters. Furthermore, the second derivatives are non-zero for all $\phi, \psi$, and the optimum hence is reached at a unique point. The objective function can be rewritten as

$$\mathbb{E}_{P \otimes Q}\left(-\phi \otimes \mathrm{I}_{\mathcal{Y}} + \mathrm{I}_{\mathcal{X}} \otimes \psi - \frac{\lambda}{e} \cdot \exp\left(\frac{-C - \phi \otimes \mathrm{I}_{\mathcal{Y}} + \mathrm{I}_{\mathcal{X}} \otimes \psi}{\lambda}\right)\right). \tag{B.9}$$

Since $P$ and $Q$ are density matrices and are constrained to lie within a compact set, there exists a compact region $\mathbb{S}$ that is independent of $P$ (but may depend on $\lambda$) such that the maximum lies inside $\mathbb{S}$. $f(P)$ can therefore be written as $f(P) = \max g(P, \phi, \psi)$, where $\phi, \psi \in \mathcal{S}$, $g$ is convex, and attains its maximum at a unique point. By Danskin's theorem [14], the result follows. $\square$

# C More Details on Quantum Wasserstein GAN

## C.1 Parameterization of the Generator

The generator $G$ is a quantum operation that maps a fixed distribution $\rho_0$ to a quantum state $P$. Two pure distributions (states with rank 1) are mapped to each other by unitary matrices. $\rho_0$ is fixed to be the pure state $\bigotimes_{i=1}^{n} e_0$. If the target state is of rank $r$, $G$ can be parameterized by an ensemble $\{(p_1, U_1), \ldots, (p_r, U_r)\}$ of unitary operations $U_i$, each of which is applied with probability $p_i$. Applying a unitary $U_i$ to $\rho_0$ produces the state $U_i \rho_0 U_i^\dagger$. Applying $G$ to $\rho_0$ thus produces the fake state $p_i U_i \rho_0 U_i^\dagger$.

Each Unitary $U_i$ is parameterized as a quantum circuit consisting of simple parameterized 1- or 2-qubit Pauli-rotation quantum gates. An $n$-qubit Pauli-rotation gate $R_\sigma(\theta)$ is given by $\exp\left(\frac{i\theta\sigma}{2}\right)$ where $\theta$ is a real parameter, and $\sigma$ is a tensor product of 1 or 2 Pauli matrices. Pauli-rotation gates can be efficiently implemented on quantum computers. Thus each unitary $U_i$ can be expressed as $U_i = \prod_j e^{\frac{i\theta_{i,j}\sigma_{i,j}}{2}}$.

## C.2 Parameterization of the Discriminator

The optimization variables in the discriminator are Hermitian operators, $\phi$ and $\psi$. There are two common parameterizations for a Hermitian matrix $H$:

1. As $U^\dagger H_0 U$, where $U$ is a parameterized unitary operator, and $H_0$ is a simpler fixed Hermitian matrix that is easy to measure. Measuring $H$ then corresponds to applying the operator $U$ and then measuring $H_0$.

2. As a linear combination $\sum_{i=0}^{\dim(H)} \alpha_i H_i$, where $H_i$s are fixed Hermitian matrices that are easy to measure. Measuring $H$ corresponds to measuring each $H_i$ to obtain the expectation value $m_i$, and then returning $\sum_{i=0}^{\dim(H)} \alpha_i m_i$ as the expected value of measuring $H$.

We choose the latter option because it allows $\xi_R$ to be conveniently approximated by a linear combination of simple Hermitian matrices. Thus $\phi$ and $\psi$ are represented by $\sum_k \alpha_k A_k$ and $\sum_l \beta_l B_l$ where $A_k, B_l$ are tensor products of Pauli matrices. The $\alpha_k$s, $\beta_l$s constitute the parameters of the discriminator.

The overall structure of the Quantum Wasserstein GAN is given in Figure 8.

Figure 7: Example parameterization of a unitary $U_i$ acting on 3 qubits. There are 12 possible 1-qubit gates and 48 possible 2-qubit gates.

Figure 8: The structure of the quantum WGAN. Here $Q$ is the input state and $\vec{e}_0$ is the $0^{\text{th}}$ computational basis vector, meaning that the corresponding system is empty at the beginning. The final gate $L$ combines the outputs of the measurements of $\phi, \psi, \xi_R$ to produce the final loss function.

## C.3 Estimating the Loss Function

The loss function is given by $\text{Tr}(Q\psi) - \text{Tr}(P\phi) - \text{Tr}((P \otimes Q)\xi_R) = \mathbb{E}_Q[\psi] - \mathbb{E}_P[\phi] - \mathbb{E}_{P \otimes Q}[\xi_R]$ where $\xi_R$ is the Hermitian corresponding to the regularizer term $\frac{\lambda}{e}\exp\left(\frac{-C - \phi \otimes I_\mathcal{Y} + I_\mathcal{X} \otimes \psi}{\lambda}\right)$.

The fake state $P$ is generated by applying a quantum operation $G$ to a fixed quantum state $\rho_0$. The quantum operation is represented by applying a set of unitary operations $\{U_1, U_2, \ldots, U_k\}$ with corresponding probabilities $\{p_1, p_2, \ldots, p_k\}$ where $k$ is the rank of the final state that would be generated:

$$P = \sum_{i \in [k]} p_i U_i \rho_0 U_i^\dagger. \tag{C.1}$$

**Lemma C.1.** *Given a quantum state $\rho = \sum_{i=1}^k \alpha_i \rho_i$ and a Hermitian matrix $H$ then $\mathbb{E}_\rho(H)$ can be estimated given only the ability to generate each $\rho_i$ and to measure $H$.*

*Proof.* Since $\rho$ is a quantum state $\{\alpha_1, \ldots, \alpha_k\}$ must form a probability distribution. Thus,

$$\mathbb{E}_\rho[H] = \text{Tr}[\rho H] = \text{Tr}\left[\sum_i \alpha_i \rho_i H\right] = \sum_i \alpha_i \text{Tr}[\rho_i H] = \sum_i \alpha_i \mathbb{E}_{\rho_i}[H] = \mathbb{E}_\alpha \mathbb{E}_{\rho_i}[H]. \tag{C.2}$$

Thus we can measure the expected value of $H$ measured on $\rho$, by sampling an $i$ with probability $\alpha_i$, measuring the expected value of $H$ on $\rho_i$, and then computing the expectation over $i$ sampled from the distribution $\alpha$. We can also simply measure the expectation value $m_i$ corresponding to each $\rho_i$ and return $\sum_i \alpha_i m_i$ as the estimate. $\square$

The unitaries $U_i$ are parameterized by a network of gates of the form $e^{i\theta_{i,j}\sigma_{i,j}}$ where $\sigma_{i,j}$ is a tensor product of the matrices $\sigma_x, \sigma_y, \sigma_z, I$ acting on some/all of the registers. With a sufficient number of such gates, any unitary can be represented by an appropriate choice of $\theta_{i,j}$. Since each $U_i$ is expressed as a composition of simple parameterized gates each of them can be implemented on a quantum computer and thus each $U_i \rho_0 U_i^\dagger$ can be generated.

Note that $P = \sum_{i \in [k]} p_i U_i \rho_0 U_i^\dagger$ and $P \otimes Q = \sum_{i \in [k]} p_i (U_i \rho_0 U_i^\dagger \otimes Q)$. From Lemma C.1, if $\phi$ and $\xi_R$ can be measured, we can estimate the terms $\mathbb{E}_P[\phi]$ and $\mathbb{E}_{P \otimes Q}[\xi_R]$. Next we show how to measure $\phi, \psi, \xi_R$ where $\phi, \psi$ are parameterized as a linear combination of tensor products of the Pauli matrices $\sigma_X, \sigma_Y, \sigma_Z, \sigma_I$.

**Lemma C.2.** *Any Hermitian that is expressed as a linear combination $\sum_i \alpha_i H_i$ of Hermitian matrices $H_i$ that can be measured on a quantum computer, can also be measured on a quantum computer.*

*Proof.* For any fixed state $\rho$,

$$\mathbb{E}_\rho[H] = \text{Tr}[\rho H] = \text{Tr}\left[\rho \sum_i \alpha_i H_i\right] = \sum_i \alpha_i \text{Tr}[\rho H_i] = \sum_i \alpha_i E_\rho[H_i]. \tag{C.3}$$

Thus each of the Hermitians $H_i$ can be separately measured and the final result is the weighted average of the corresponding expectation values with coefficients $\alpha_i$.

If the $\alpha_i$ form a probability distribution, the expectation can be estimated by sampling a batch of indices from the distribution of $\alpha_i$, measuring $H_i$, and estimating the expectation averaging over the sampled indices. This procedure can be more efficient if some of the $\alpha_i$ are of very small magnitude in comparison to the others. Note that any Hermitian that can be written by as a linear combination $\sum_i \beta_i H_i$ where each $H_i$ is easy to measure can be transformed such that the coefficients form a probability distribution as $(\sum_i |\beta_i|) \sum_i \frac{|\beta_i|}{\sum_i |\beta_i|} \text{sgn}(\beta_i) H_i$. If $H_i$ can be measured on a quantum computer, $-H_i$ can also be measured by measuring $H_i$ and negating the result. $\square$

Tensor products of Pauli matrices can be measured on quantum computers using elementary techniques [28]. As a result, Lemma C.2 implies that $\phi, \psi$ can be measured on a quantum computer.

Now, we prove the following lemma for expressing the regularizer term $\xi_R$:

**Lemma C.3.** *The Hermitian corresponding to the regularizer term $\xi_R$ can be approximated via a linear combination of Hermitians from $\{\Sigma, \text{SWAP} \cdot \Sigma\}$ where $\Sigma$ is a tensor product of 2-dimensional Hermitian matrices.*

*Proof.* Since $C = \frac{\text{I}-\text{SWAP}}{2}$,

$$\exp\left(\frac{-C - \phi \otimes \text{I}_\mathcal{Y} + \text{I}_\mathcal{X} \otimes \psi}{\lambda}\right) = \exp\left(\frac{\text{SWAP} - \text{I} - 2\phi \otimes \text{I}_\mathcal{Y} + 2\text{I}_\mathcal{X} \otimes \psi}{2\lambda}\right). \tag{C.4}$$

Observe the following two facts:

- if $\Sigma_1$ and $\Sigma_2$ are both tensor products of 2-dimensional Hermitian matrices, then $\Sigma_1 \cdot \Sigma_2$ is also a tensor product of 2-dimensional Hermitian matrices;
- if $\Sigma$ is a tensor product of 2-dimensional Hermitian matrices, then $\text{SWAP} \cdot \Sigma \cdot \text{SWAP}$ is also a tensor product of 2-dimensional Hermitian matrices.

As a result, any integral power of $\text{SWAP} - \text{I} - 2\phi \otimes \text{I}_\mathcal{Y} + 2\text{I}_\mathcal{X} \otimes \psi$ can be written as a linear combination of the matrices $\{\Sigma, \text{SWAP} \cdot \Sigma\}$ where $\Sigma$ is a tensor product of 2-dimensional Hermitian matrices. Thus any Taylor approximation of $\exp(\text{SWAP} - \text{I} - 2\phi \otimes \text{I}_\mathcal{Y} + 2\text{I}_\mathcal{X} \otimes \psi)$ is a linear combination of the same Hermitian matrices, each of which can be easily measured on a quantum computer. Thus the Taylor series for the exponential can be used to approximately measure the regularizer term.

A representation as a linear combination of the Hermitians $\{\Sigma, \text{SWAP} \cdot \Sigma\}$, where $\Sigma$ is a tensor product of Pauli matrices, can be obtained more easily for a relaxed regularizer term

$$\xi'_R = \exp\left(\frac{-C}{2\lambda}\right) \exp\left(\frac{-\phi \otimes \text{I}_\mathcal{Y} + \text{I}_\mathcal{X} \otimes \psi}{\lambda}\right) \exp\left(\frac{-C}{2\lambda}\right); \tag{C.5}$$

this is motivated by the Trotter formula [41] of matrix exponentiation: for any Hermitian matrices $A, B$ such that $\|A\|, \|B\| \leq \delta \leq 1$, $\|e^{A+B} - e^A e^B\| = O(\delta^2)$ but $\|e^{A+B} - e^{A/2}e^B e^{A/2}\| = O(\delta^3)$. Using this regularizer gives us a concrete closed form for $\xi'_R$ as a linear combination of simpler Hermitian matrices. It is less computationally intensive to compute than the original regularizer, since the only operation acting on $2n$ qubits at the same time is SWAP. This relaxation also yields good numerical results in practice.

Since $(-\phi \otimes \text{I}_\mathcal{Y})(\text{I}_\mathcal{X} \otimes \psi) = (\text{I}_\mathcal{X} \otimes \psi)(-\phi \otimes \text{I}_\mathcal{Y}) = (-\phi \otimes \psi)$, the central term in the RHS of (C.5) is an exponential of commuting terms. If $A$ and $B$ are commuting matrices, we have $\exp(A + B) = \exp(A)\exp(B)$, and hence

$$\xi'_R = \exp\left(\frac{-C}{2\lambda}\right) \exp\left(\frac{-\phi}{\lambda}\right) \otimes \exp\left(\frac{\psi}{\lambda}\right) \exp\left(\frac{-C}{2\lambda}\right). \tag{C.6}$$

We choose $\phi$ and $\psi$ to be tensor products of terms of the form $a\sigma_x + b\sigma_y + c\sigma_z + d\text{I}$. It can be verified that $\sigma_i \sigma_i = \text{I}$ and $\sigma_i \sigma_j + \sigma_j \sigma_i = 2\delta_{i,j}\text{I}$ and therefore $(a\sigma_x + b\sigma_y + c\sigma_z)^2 = (a^2 + b^2 + c^2)\text{I}$. Given $r = \bigotimes_{i=1}^n (a_i\sigma_x + b_i\sigma_y + c_i\sigma_z + d_i\text{I})$, we therefore have $r^2 = \bigotimes_{i=1}^n \left(d_i(a_i\sigma_x + b_i\sigma_y + c_i\sigma_z + d_i\text{I}) + \Pi_{i=1}^n(a_i^2 + b_i^2 + c_i^2 + d_i^2)\text{I}\right)$ and therefore by induction,

$$r^k = \bigotimes_{i=1}^n \left( d_i^{k-1}(a_i\sigma_x + b_i\sigma_y + c_i\sigma_z + d_i\text{I}) + \left(\sum_{j=0}^{k-2} d_i^j\right)(a_i^2 + b_i^2 + c_i^2 + d_i^2)\text{I} \right). \tag{C.7}$$

Eq. (C.7) can be used to expand $\exp(-\phi/\lambda) \otimes \exp(\psi/\lambda)$ using the truncated Taylor series for the exponential. Thus $\exp(-\phi/\lambda) \otimes \exp(\psi/\lambda)$ can be approximated by a linear combination of gates in $\Sigma$ up to any desired accuracy.

In addition, $C = \frac{\text{I}-\text{SWAP}}{2}$ implies that $C$ is a projector, i.e., $C^k = C$ for all $k \in \mathbb{N}^*$ and $C^0 = \text{I}$. This can be used to express $\exp(C)$ in terms of only $\text{I}$ and $C$:

$$\exp\left(\frac{-C}{2}\right) = \text{I} + \sum_{j=1}^\infty \frac{C}{(-2)^j j!} = \text{I} + \left[\exp\left(\frac{-1}{2}\right) - 1\right]C. \tag{C.8}$$

Using (C.7) and (C.8) we can compute an approximate expression (with any desired accuracy) for the relaxed regularizer $\xi'_R$ as a linear combination of the Hermitian $\{\Sigma, \text{SWAP} \cdot \Sigma\}$ where $\Sigma$ is a tensor product of Hermitian matrices. $\qquad \square$

Finally from Lemma C.1 Lemma C.2 Lemma C.3, each of the terms $\mathbb{E}_Q[\psi], \mathbb{E}_P[\phi], \mathbb{E}_{P \otimes Q}[\xi_R]$ can be computed on a quantum computer.

## C.4 Direct Estimation of Gradients

In this subsection, we show how the gradients with respect to the parameters of the qWGAN can be directly estimated using quantum circuits. Suppose we have the following parameterization for the optimization variables:

$$\rho_0 = \bigotimes_{i=1}^{d} \vec{e}_0 \vec{e}_0^{\dagger}, \qquad P = \sum_{i=1}^{r} p_i U_i \rho_0 U_i^{\dagger}, \qquad U_i = \prod_j e^{\frac{i\theta_{i,j}H_{i,j}}{2}} \qquad (\text{C.9})$$

and

$$\phi = \sum_k \alpha_k A_k, \qquad \psi = \sum_l \beta_l B_l, \qquad (\text{C.10})$$

where $H_j, A_k, B_l$ are tensor products of Pauli matrices. The parameters of the generator are given by the variables $p_i, \theta_{i,j}$ and the parameters of the discriminator are given by $\alpha_k, \beta_l$. As shown in Lemma C.3, the regularizer term $R$ can be written as $\sum_q r_q R_q$ where each $R_q$ is either a tensor product of Pauli matrices or a product of SWAP with a tensor product of Pauli matrices. Thus the loss function is given by

$$L = \text{Tr}[Q\psi] - \text{Tr}[P\phi] - \text{Tr}\left[(P \otimes Q)R\right], \qquad (\text{C.11})$$

and hence

$$\frac{\partial L}{\partial p_i} = -\text{Tr}[U_i \vec{e}_0 \vec{e}_0^{\dagger} U_i^{\dagger} \phi] - \text{Tr}\left[(U_i \vec{e}_0 \vec{e}_0^{\dagger} U_i^{\dagger} \otimes Q)R\right]. \qquad (\text{C.12})$$

To compute the partial derivative with respect to the parameters $p_i$, we create a fake state using only the unitary $U_i$, and compute the regularizer term as shown before:

$$\frac{\partial L}{\partial \alpha_k} = -\text{Tr}[PA_k] - \text{Tr}\left[(P \otimes Q)\frac{(A_k \otimes I_{\mathcal{Y}})R}{\lambda}\right]; \qquad (\text{C.13})$$

$$\frac{\partial L}{\partial \beta_l} = \text{Tr}[QB_l] - \text{Tr}\left[(P \otimes Q)\frac{(I_{\mathcal{X}} \otimes B_l)R}{\lambda}\right]. \qquad (\text{C.14})$$

Clearly $(A_k \otimes I_{\mathcal{Y}})R$ and $(I_{\mathcal{X}} \otimes B_l)R$ can be written as linear combinations of products of SWAP and tensor products of Pauli matrices, because such form exists for $A_k, B_l, R$. Thus these gradients can be measured as shown in Lemma C.2.

Regarding the gradients with respect to $\theta_{i,j}$, we have

$$\frac{\partial L}{\partial \theta_{i,j}} = \frac{\partial \text{Tr}[\phi(U_i \rho_0 U_i^{\dagger})]}{\partial \theta_{i,j}} - \frac{\partial \text{Tr}[\xi_R(U_i \rho_0 U_i^{\dagger} \otimes Q)]}{\partial \theta_{i,j}}. \qquad (\text{C.15})$$

The terms $\frac{\partial \text{Tr}[\phi(U_i \rho_0 U_i^{\dagger})]}{\partial \theta_{i,j}}, \frac{\partial \text{Tr}[\xi_R(U_i \rho_0 U_i^{\dagger} \otimes Q)]}{\partial \theta_{i,j}}$ can be evaluated by modifying the quantum circuits for $U_i$ using with an ancillary control register, using previously known techniques [36, Section III. B]. This allows us to evaluate the partial derivatives of the loss function w.r.t. the $\theta_{i,j}$ parameters.

## C.5 Computational Cost of Evaluating the Loss Function

Consider a quantum WGAN designed to learn an $n$-qubit target state with rank $r$; the generator hence consists of $r$ unitary matrices. Suppose that each unitary $U_i$ is a composition of at most $N$ fixed unitary gates. Furthermore, assume that $\phi$ and $\psi$ are parameterized as a linear combination of at most $M$ tensor products of Pauli matrices. The size of the network (the number of parameters) is thus $O(rNM)$.

The loss function consists of 3 terms:

- The expectation value of $\phi$ measured on the state $P$.
- The expectation value of $\psi$ measured on the state $Q$.
- The expectation value of $\xi_R$ measured on the state $P \otimes Q$.

The complexity of a quantum operation is quantified by the number of elementary gates required to be performed on a quantum computer. We show that a single measurement of $\phi$ on $U_i \rho_0 U_i^{\dagger}$, $\psi$ on $Q$, and $\xi_R$ on $U_i \rho_0 U_i^{\dagger} \otimes Q$ can be carried out using $\text{poly}\left(n, k, N, M, \log\left(\frac{1}{\epsilon}\right)\right)$ gates.

The expectation values can then be estimated by computing the empirical expectation on a batch of measurements. These expectation values are combined as shown earlier in Supplemental Materials C.3 to obtain the expected values measured on $P$ and $P \otimes Q$.

First, $\xi_R$ can be approximated to precision $\epsilon$ via truncation of a Taylor series consisting of $\log\left(\frac{1}{\epsilon}\right)$ terms. Thus $\xi_R$ is approximated by a linear combination of $\mathrm{poly}\left(M, \frac{1}{\epsilon}\right)$ fixed Hermitian matrices of the form $\Sigma$ or $\mathrm{SWAP} \cdot \Sigma$ where each $\Sigma$ is a tensor product of 2-dimensional Hermitian matrices.

Second, by the Solovay-Kitaev theorem [15], any $n$-qubit unitary operator can be implemented to precision $\epsilon$ using $\mathrm{poly}\left(\log\left(n, \frac{1}{\epsilon}\right)\right)$ gates. Similarly, any fixed $n$-qubit Hermitian matrix can be measured using a circuit with $\mathrm{poly}\left(n, \log\left(\frac{1}{\epsilon}\right)\right)$ gates. Consequently:

- $\psi$ can be measured on $Q$ using $M$ measurements of fixed tensor products of Pauli matrices, therefore using $\mathrm{poly}\left(n, M, \log\left(\frac{1}{\epsilon}\right)\right)$ gates.

- $\phi$ can be measured on $U_i \rho_0 U_i^\dagger$ for any $i$ using $M$ measurements of fixed tensor products of Pauli matrices, therefore using $\mathrm{poly}\left(n, M, \log\left(\frac{1}{\epsilon}\right)\right)$ gates.

- $\xi_R$ can be measured on $U_i \rho_0 U_i^\dagger \otimes Q$ for any $i$ using $\mathrm{poly}\left(M, \frac{1}{\epsilon}\right)$ measurements of fixed tensor products of Pauli matrices, therefore using $\mathrm{poly}\left(n, M, \log\left(\frac{1}{\epsilon}\right)\right)$ gates.

- Each unitary $U_i$ can be applied by a composition of $N$ fixed unitaries, therefore using $\mathrm{poly}\left(n, N, \log\left(\frac{1}{\epsilon}\right)\right)$ gates.

From Supplemental Materials C.4, it can be seen that the partial derivatives with respect to the parameters $p, \alpha, \beta$ are each computed by the same procedure as the loss function with some of the variables restricted. Furthermore, the partial derivatives with respect to $\theta_{i,j}$ can be evaluated using the circuit for $U_i$ with an ancillary register and a constant number of extra gates [36]. Each partial derivative therefore has the same complexity as the loss function. Since there are $O(rNM)$ parameters, the total gradient can be evaluated with a multiplicative overhead of $O(rNM)$ compared to evaluating the loss function.

## D  More Details on Experimental Results

**Pure states**  We used the quantum WGAN to learn pure states consisting of $1, 2, 4,$ and $8$ qubits. In this case, the generator is fixed to be a single unitary. The parameters to be chosen in the training are $\lambda$ (the weight of the regularizer) and $\eta_g, \eta_d$ (the learning rates for the discriminator and generator parameters, respectively). The training parameters for our experiments for learning pure states are listed in Table 1.

| Parameters | 1 qubit | 2 qubits | 4 qubits | 8 qubits |
|---|---|---|---|---|
| $\lambda$ | 2 | 2 | 10 | 10 |
| $\eta = \eta_g = \eta_d$ | $10^{-1}$ | $10^{-1}$ | $10^{-1}$ | $10^{-2}$ |

Table 1: Parameters for learning pure states.

For 1,2, and 4 qubits, in addition to Figure 3, we also plot the average loss function for a number of runs with random initializations in Figure 9 which shows the numerical stability of our quantum WGAN.

**Mixed states**  We also demonstrate the learning of mixed quantum states of rank 2 with $1, 2,$ and $3$ qubits in Figure 4. The generator now consists of 2 unitary operators, and 2 real probability parameters $p_1, p_2$ which are normalized to form a probability distribution using a softmax layer. The learning rate for the probability parameters is denoted by $\eta_p$. The training parameters are listed in Table 2.

| Parameters | 1 qubits | 2 qubits | 3 qubits |
|---|---|---|---|
| $\lambda$ | 10 | 10 | 10 |
| $\eta_d, \eta_g, \eta_p$ | $(10^{-1}, 10^{-1}, 10^{-1})$ | $(10^{-1}, 10^{-1}, 10^{-1})$ | $(10^{-1}, 10^{-1}, 10^{-1})$ |

Table 2: Parameters for learning mixed states.

Figure 9: Average performance of learning pure states (1, 2, 4 qubits) where the black line is the average loss over multi-runs with random initializations and the shaded area refers to the range of the loss.

**Learning pure states with noise**  In a recent experiment result [48], a quantum-classical hybrid training algorithm using the KL divergence between classical measurement outcomes as the loss function on the canonical Bars-and-Stripes data set was performed on an ion-trap quantum computer. Specifically, they use the generator in Figure 10. Even though the goal of [48] is to generate a classical distribution, we still deem it as a good example of practically implementable quantum generator to testify our quantum WGAN.

Figure 10: The generator circuit used in Ref. [48] where $Z$ stands for the $e^{i\theta\sigma_z}$ gate, $X$ stands for the $e^{i\theta\sigma_x}$ gate, and $XX$ stands for the $e^{i\theta\sigma_x\otimes\sigma_x}$ gate.

We use the same training parameters as in the noiseless case (Table 1). Furthermore, we add the sampling noise (modeled as a Gaussian distribution with standard deviation $\sigma$) which is a reasonable approximation of the noise for the ion-trap machine [47]. Our results show that the quantum WGAN can still learn a 4-qubit mixed state in the presence of this kind of noise. As is to be expected, noise with higher degrees (i.e., higher $\sigma$) increases the number of epochs required before the state is learned successfully. The corresponding results are plotted in Figure 5.

Our finding also demonstrates the different outcomes between choosing different metrics as the loss function. In particular, some of the training results reported in [48] demonstrate a KL distance $< 10^{-4}$ but the actual quantum fidelity is only about $0.16$. On the other side, our quantum WGAN is guaranteed to achieve close-to-1 fidelity all the time.

**Application: Approximating Quantum Circuits**  The quantum Wasserstein GAN can be used to approximate the behavior of quantum circuits with many gates using fewer quantum gates. Consider a quantum circuit $U_0$ over $n$ qubits. It is well known [28] that there exists an isomorphism between $n$

qubit quantum circuits $U$ and quantum states $\Psi_U$ such that

$$\Psi_U = \frac{1}{\sqrt{2^n}} \sum_{i=0}^{2^n-1} (U \otimes \mathrm{I})(\vec{e}_i \otimes \vec{e}_i) = \frac{1}{\sqrt{2^n}} \sum_{i=0}^{2^n-1} (U(\vec{e}_i) \otimes \vec{e}_i). \tag{D.1}$$

The quantum Wasserstein GAN can be used to learn a smaller quantum circuit $U_1$ such that $\Psi_{U_1}$ is close to $\Psi_{U_0}$. This can be done by setting the real state to $\Psi_{U_0}$, and using the GAN to learn to generate it using a circuit of the form $(U_1 \otimes \mathrm{I})$ applied to $\frac{1}{\sqrt{2^n}} \sum_{i=0}^{2^n-1} (\vec{e}_i \otimes \vec{e}_i)$. The fidelity between $\Psi_{U_1}$ and $\Psi_{U_0}$ is given by the average output fidelity for uniformly chosen inputs to $U_1$ and $U_0$.

We apply these techniques to the quantum circuit that simulates the evolution of a quantum system in the 1-dimensional nearest-neighbor Heisenberg model with a random magnetic field in the $z$-direction (considered in [11]). The time evolution for time $t$ is described by the unitary operator $e^{i\hat{H}t}$ with the Hamiltonian $\hat{H}$ given by

$$\hat{H} = \sum_{j=1}^{n} \left( \sigma_x^{(j)} \sigma_x^{(j+1)} + \sigma_y^{(j)} \sigma_y^{(j+1)} + \sigma_z^{(j)} \sigma_z^{(j+1)} + h^{(j)} \sigma_z^{(j)} \right) \tag{D.2}$$

where $\sigma_i^{(j)}$ denotes the Pauli gate $\sigma_i$ applied at the $j^{th}$ qubit, and the $h^{(j)} \in [-h, h]$ are uniformly chosen at random.

We study the specific case with $t = n = 3$ and $h = 1$, with a fixed target error of $\epsilon = 10^{-3}$ in the spectral norm. Quantum circuits for simulating Hamiltonians that are represented as the sum of local parts, $e^{iHt} = e^{it \sum_{i=1}^{L} \alpha_j H_j}$, are obtained using $k^{th}$ order Suzuki product formulas $S_{2k}$ defined by

$$S_2(\lambda) = \prod_{j=1}^{L} \exp(\alpha_j H_j \lambda/2) \prod_{j=L}^{1} \exp(\alpha_j H_j \lambda/2) \tag{D.3}$$

$$S_{2k}(\lambda) = [S_{2k-2}(p_k\lambda)]^2 S_{2k-2}((1-4p_k)\lambda)^2 [S_{2k-2}(p_k\lambda)]^2 \tag{D.4}$$

where $p_k = 1/\left(4 - 4^{1/(2k-1)}\right)$ for $k \geq 1$.

We then approximate $e^{iHt}$ by $\left[ S_{2k}\left(\frac{it}{r}\right) \right]^r$. Obtaining error $\epsilon$ in the spectral norm requires $r = \frac{(Lt)^{1+1/2k}}{\epsilon^{1/2k}}$. From (D.3), each evaluation of $S_{2k}$ requires $(2L)5^{k-1}$ gates of the form $e^{iH_j\theta}$ where $\theta$ is a real parameter. In the case of the Hamiltonian (D.2), it is the sum of 12 terms each of which is the product of up to 2 Pauli matrices. Thus the $k^{th}$ order formula $S_{2k}$ yields a circuit for simulating (D.2) requiring $(24)5^{k-1} \frac{(36)^{1+1/2k}}{0.001^{1/2k}}$ gates of the form $e^{i\theta\sigma}$ where $\sigma$ is a product of up to 2 Pauli matrices. These are the gates used in the parameterization of our quantum Wasserstein GAN, and can be implemented easily on ion trap quantum computers. The smallest circuit is obtained using $S_2$ and requires $\sim 11900$ gates.

Using the quantum Wasserstein GAN for 6-qubit pure states, we discovered a circuit for the above task with 52 gates, an average output fidelity of 0.9999, and a worst case error 0.15. The worst case input is not realistic, and thus the 52 gate circuit provides a very reasonable approximation in practice.

## Footnotes

[3]Uniform circuits have equivalent computational power as Turing machines; however, they are more convenient to use in quantum computation.