[Reviews · NeurIPS 2019]

Reviewer 1



After rebuttal: Thank you for the rebuttal. It helped me understand the sampling more / evaluating the loss more. Also, as your scheme is not designed to generalize OT to the quantum setting, I am fine that the quantum Wasserstein semimetric does not allow for a general cost function. Based on these and the promising real life experiment mentioned in the rebuttal, I have decided to raise my review to marginally above the acceptance rate. ------------------------------------------------------------------------------------- The paper introduces the Wasserstein semimetric between quantum states, which is then applied in learning to generate an empirical quantum state. The properties required for a semimetric are shown and furthermore the authors show that it behaves in a smooth way with respect to the quantum states. The method is then compared to other work in the literature in a non-direct way, as the referenced work have carried out the experiments in very different settings. The paper is well written and the contributions are introduced in a clear way mathematically. For a non-expert in quantum computing the paper was a hard read though, mostly due to the extensive background needed (as evidenced by the 20+ page supplementary material). On top of this, in many cases there is a reference to the supplementary material for discussions and analysis, which I think harms the ability of this paper to stand out on its own. To some extend this is fine, but in this case I think it made the paper lack important content. I was missing more comparison between the classical WGANs and qWGANs. For example, it remains unclear to me how the sampling is done in the qWGAN case. Furthermore, I was not able to find how to evaluate the cost function c(x,y). In the classical WGAN case, x and y are samples from distributions, but what are they in this quantum case? Also, which cost function is used in this work? As far as I understand, the matrix C does not depend on the supports of P and Q ( that is, there is no ‘c’), which is very different to the classical situation. As C does not depend on a cost function on the underlying ‘sample space’, whichever that is in the quantum case, it is difficult to agree that (4.1) would be a natural generalization of the primal formulation of the Wasserstein metric. This is because the fundamental property of optimal transport is the ability to take the geometry of the sample space into account. Specific remarks: - Line 30: seminar -> seminal? - Line 58-59: KL and JS divergences aren’t metrics in the strict sense of the word. - Line 67: The main contribution is claimed to be the initiation of the study of quantum Wasserstein GANs. This wording sounds quite grandiosa, taken into account that GANs have been studied in the quantum setting already. - Line 75: The references for the formulation of the Wasserstein sense in the coupling form (primal) and ‘optimal transport’ form (dynamic I guess?) are really vague, especially as Villani’s book discusses all three formulations (primal, dual and dynamic). - Line 118-119: Denoting the distributions and densities with the same symbols is a bit confusing, although understandable as this notation is not that present in rest of the work. - Line 132-133: Weight clipping was the heurestic used in the vanilla WGAN, but is not the state-of-the-art anymore (thus the word often is a bit off here). I believe the gradient penalty method is much more prevalent nowadays. - Line 162-163: ‘A more reasonable choice of C would be the projection onto the orthogonal…’, this was a bit confusing at first, but I guess here the projection of the cost matric c(x,y) is meant? - Line 194-195: The terminology of ‘expected value of observing Hermitian psi on quantum state Q’ is not familiar to me. Does this have a physical interpretation? The probability of observing something sounds more familiar. - Line 209-210: What is a ‘simple tensor product of Pauli matrices’? - Line 211-212: the ’s’ after the expectation is confusing. Similarly the s after alpha_k and beta_l, avoid mixing mathematical notation with ‘grammar notation’. - Eq. (4.4): I guess min here should be with respect to the parameters of G. - Line 239: Using notation like \{p_i\}_{i=1}^N instead of just \{p_i\} would be clearer. - Experimental section: I believe fidelity is only defined in the supplementary material, but the right place for this would be in the main paper in my opinion. It might also be interesting, from the optimal transport point of view, to remark that the fidelity is closely related to the 2-Wasserstein metric between SPD matrices. -Supplementary material: in the proof of Theorem 1. 3), C is defined as (I - SWAP)/2, but in 3) C = (I + SWAP)/2 is used (or atleast written down).

Reviewer 2



This work connects together the idea of using Wasserstein distance for cost functions in GANs to stabilize training with the recently introduced notion of quantum GANs. To someone with sufficient visibility into both the machine learning and quantum computing literature, this might be seen as a natural progression and tying together of ideas from existing works in a straightforward way. Admittedly, though, there may be very few such dual-field experts out there. For quantum computing researchers, this work provides a method to potentially train larger-scale models on noisy machines than have been previously managed. To machine learning researchers, this work could be interesting because extends the familiar notion of WGANs to run on a new type of (available) hardware that can potentially model interesting intractable distributions. The paper is technically clear. I did not spot any obvious flaws or scientific errors. The numerical results support the motivation and theoretical results of the paper (that training should be smooth and efficient). The submission is clearly written and the argumentation is clear. I could follow the narrative of the work without any big issues, and extensive supporting material, proofs, and code are provided in the appendix for those seeking more concrete details. There were a few minor typos (more of a nuisance than any significant barrier to understanding). I would recommend that the authors go through an additional round of editing to polish it up. This work builds upon previous work in a positive way, showing how existing training methods for quantum GANs can be made smoother and more robust, potentially allowing much larger models to be trained easier on noisy hardware. Regarding significance, I would vote in favour of acceptance of this paper in NeurIPS because it is important for ML researchers to be made aware of advances and potential advantages of near-term quantum computing devices for machine learning. In particular, the ability to build generative models which can model and sample from otherwise intractable distributions---with the ability to run these models on currently available (or near-term) noisy quantum hardware---might be of general interest to the machine learning community and should be highlighted. Wider awareness of the current ideas in quantum machine learning could potentially lead to interesting breakthroughs and new bridges being built by experts from both sides.

Reviewer 3



While it is challenging to me to understand all the new properties of the proposed quantum Wasserstein GAN, The motivation and background knowledge on the Wasserstein distance and Wasserstein GAN are presented clearly. The idea of applying the Wasserstein distance and its derived GAN model on quantum data sounds interesting and straightforward. It gives me the impression that all the used components including the regularization are studied well on regular data, and the paper seems to apply them to quantum data with some special designs. The experiments look very insufficient. It only evaluates the proposed method on numerical experiments without any comparisons against relevant methods or their adaptation.

Reviewer 4



I think the claim that the proposed definition of qW is new is not true: - This definition was considered before by the team of Golse and collaborators [1], and is recently reviewed in [2]. These authors consider an apparently related but different cost function C and in particular introduce an epsilon parameter that should goes epsilon->0 to obtain a definite positive semi-distance, but my feeling is that it is because these work consider possibly infinite dimensional Hilbert spaces (see below about why I think this is important). - The same idea of using partial trace constraints was also considered in [3]. This work also considers an extra spatial dimension, but for a space with a single point, this gives the same type of problem. - The idea of using quantum entropic regularization is considered in [4]. This is for a slightly different problem, where partial trace constraints are replaced by quantum-KL discrepancy ("unbalanced qOT"), but this leads to the same expm Gibbs factorization of the solution. Regarding the importance of using qW with respect to simpler losses. The authors claim that "the discontinuity of ... the trace distance", but I think this is not true, the trace distance is continuous. So, to me, since the paper only considers finite dimensional settings, it is not clear how to support the narrative that qW is better than alternative distances (just as with classical OT, where one has to goes to continuous spaces to see the difference in term of induced topology between TV and W). So, the work of Golse and collaborators is very relevant here, since they do provide a quantitative comparison between qW and W applied to Husimi transforms of the states. The present paper lacks this type of control to justify the use of the qW loss. Related to this, it is a pity that, despite not having theoretical results regarding their qW loss, they did neither provide numerical results supporting the superiority when training GANs. The nuclear norm and the Bures metric (as review in the supplementary section "Distance measure") are both convex functionals, so one could also dualize and develop a GAN, just as vanilla GAN does for the JS divergence. It is important to realize that by modifying the dual problem and using a neural network actually stabilize the primal divergence, and in practice, vanilla GAN is still the mostly used method, and the superiority of the WGAN is only very marginal (or even not true). Also, and this is related to these issues, I think the statement that the qW loss is differentiable is wrong (just as W is not differentiable). To have differentiability, the solution of the dual problem must be unique (up to an additive constant). This is in general not true, just as for classical OT. This is why adding entropic regularization is important, to obtain a smooth loss function. It is a shame that the authors of the paper did not insist on this (in my mind) important contribution, since smoothing the W distance is a very important topic, especially when dealing with high dimensional problem where OT is known to perform very badly (curse of dimensionality). Lastly, my overall feeling is that the paper does a poor job at actually presenting the method and the take home message. Most of the interesting part is actually in the appendix, in particular: - the precise definition of the cost C is super important, and should be given in the main text *before* stating the theory. One should not have to read the supplementary to be able to understand the statement of the theorem (of course it is fine with me to have the *proofs* in the supplementary, but not the hypotheses). - all the narrative and intuitive explanation about why qW is better than other norms is super important. Currently, this material is spread in the supplementary, and is explained in a quite fuzzy way. In particular, the paragraph "Classical examples of simple sequence ..." is very important, but there is no equation and no precise statement (in particular in light of my previous remark about the fact that the nuclear norm is continuous just as qW). To summarize, I have mixed feeling about this paper, but I would still support acceptance under the conditions that: - The authors give a proper credit to previous works on static Kantorovitch-type formulations for qOT, and in particular put in context their main contribution with respect to the previous work of Golse and collaborators (the critical part being to understand the differences/similarities between the cost functions, and understand this in the context where the dimension -> +infty). - Does some "mass transport" between the supplementary and the main text so that the paper becomes self-contained and explains more clearly the advantage of the proposed metric. ---- References --- [1] On the Mean-Field and Classical Limits of Quantum, F. Golse, C. Mouhot, T. Paul, Mechanics, Commun. Math. Phys. 343 (2016), 165–205. [2] Quantum optimal transport is cheaper Emanuele Caglioti, François Golse, Thierry Paul https://arxiv.org/abs/1908.01829v1 [3] Matrix-valued Monge-Kantorovich Optimal Mass Transport Lipeng Ning, Tryphon T. Georgiou, Allen Tannenbaum, https://arxiv.org/abs/1304.3931 [4] Quantum Optimal Transport for Tensor Field Processing Gabriel Peyré, Lenaïc Chizat, François-Xavier Vialard, Justin Solomon, https://arxiv.org/abs/1612.08731

[Author Response · NeurIPS 2019]

**Response To Reviewer #2.**

**Running on real-world quantum hardware:** We note that publicly available machines are less powerful for useful

demonstrations mainly due to their size limit (# of gates and qubits). To include the real-world noise model in our

simulations, in lines 271-278, we describe exactly the same type of *noisy* simulation from one ion-trap group.

**Real-world applications of proposed quantum WGAN.** In the revised version of the paper, we will add a real-world

application of the quantum WGAN suggested by Reviewer 2. The specific task is to approximately implement large

quantum circuits (denoted by $U_0$) by smaller ones (denoted by $U_1$). The connection is as follows: to approximate $U_0$ on,

e.g., $|0\rangle$, quantum WGAN can find a more succinct generator $U_1$ s.t. $U_1 |0\rangle \approx U_0 |0\rangle$. To approximate on all inputs, we

use the quantum state-channel isomorphism (i.e.,the Choi-Jamiołkowski state), which is $|\Psi_0\rangle = U_0 \otimes I |\Phi\rangle$ where $|\Phi\rangle$

is the maximally entangled state. It suffices to find a more succinct generator $U_1$ such that $|\Psi_1\rangle = U_1 \otimes I |\Phi\rangle \approx |\Psi_0\rangle$.

The fidelity between $|\Psi_0\rangle$ and $|\Psi_1\rangle$ then becomes the average output fidelity over uniformly chosen inputs to $U_0/U_1$.

Specifically, we studied the quantum Hamiltonian simulation circuit for 1-d 3-qubit Heisenberg model (in Eqn. (1) of

arXiv:1711.10980v1). The best-known quantum circuit with the worst case error $10^{-3}$ (in operator norm) has over

11,900 gates. Using the above approach and our quantum WGAN (for 6-qubit), we discovered a circuit $U_1$ with 52

gates with an average output fidelity over 0.9999 and a worst-case error 0.15. The worst-case input is not realistic in

current experiments and hence the high average fidelity implies very reasonable approximation in practice. This task

could only be achieved using our quantum WGAN, rather than previous quantum GAN proposals, given its complexity.

**Response To Reviewers #1 and #3.**

**Differences between classical and quantum data/sampling.** We want to emphasize that the quantum extension of

WGAN was *not* a straightforward extension of WGAN as suggested by Reviewer 3, due to the essential difference

between quantum and classical data. Consider a classical random bit $b$ with density $(0.4(b = 0), 0.6(b = 1))$. A

classical readout (or sample) refers to a random variable with this distribution. In quantum mechanics, these are two

*separate* concepts. An operator extension of density, called the *density* operator (semidefinite operators with trace 1,

lines 135-150), represents an ensemble of *quantum data*, which includes information of both pure quantum states (as

unit vectors) and their density. A classical readout on quantum states refers to a *quantum measurement* (lines 439-449).

When measuring density operator $Q$ using observable $\psi$, its outcome is a random variable with expectation $\mathrm{Tr}(Q\psi)$.

Classical random bit $(0.4, 0.6)$ is simply a $\mathrm{diag}(0.4, 0.6)$ density operator and there is only one allowed measurement

in classical mechanics. Hence, there is no distinction between these two concepts for classical data. A quantum bit

(qubit) refers to a $2 \times 2$ density operator with potentially complicated off-diagonal terms. Moreover, one can have many

measurements for one quantum data. This justifies why density operators represent the entity of quantum data.

The outcome of a quantum generator must hence be mathematically represented by a single density operator. A classical

random bit can also be represented by a diagonal density operator, although it might not be very intuitive in the first use.

**Cost function and the geometry of the sample space in qWGAN.** The definition of cost function for quantum data

must work with density operators. Let us first formulate the classical cost function (2.1) in the density operator

form. Consider one random bit and choose c(0,0)=c(1,1)=0 and c(0,1)=c(1,0)=1. Then (2.1) becomes $\sum_{a,b\in\{0,1\}}$

$\pi(a,b)c(a,b)$ where $\pi$ is the coupling of two random bits, which is mathematically the same as $\mathrm{Tr}(\pi C)$ where $\pi =$

$\mathrm{diag}(\pi(0,0), \pi(0,1), \pi(1,0), \pi(1,1))$ and $C = \mathrm{diag}(c(0,0), c(0,1), c(1,0), c(1,1))$. (Note $C$ is independent of $\pi$.)

Our (3.1) is the quantum extension of the above with important distinctions. In (3.1), $\pi$ is a density operator for the

quantum coupling of $P$ and $Q$, with potentially very complicated off-diagonal terms. The diagonal $C$ in the classical

case does not work for off-diagonal $\pi$. It is easy to find examples of $P$ such that $\mathrm{qW}(P, P) > 0$ with the diagonal $C$.

Our solution is to leverage the concept of *symmetric subspace* in quantum information. The projection onto any

subspace $V$ is a matrix with eigenspace $V$ with eigenvalue 1, and eigenspace $V^\perp$ with eigenvalue 0. The projection

onto the symmetric subspace, denoted $\Pi_{\mathrm{sym}}$=(I+SWAP)/2, has the property that $\Pi_{\mathrm{sym}}P \otimes P = P \otimes P$. By choosing $C$

to be the projection of its orthogonal subspace, i.e., C=I- $\Pi_{\mathrm{sym}}$ =(I-SWAP)/2, we have $\mathrm{qW}(P, P) = 0$ for any $P$.

It also encodes the geometry of the space of quantum states. Choose P=$\vec{v}\vec{v}^\dagger$ and Q=$\vec{u}\vec{u}^\dagger$ and $\mathrm{Tr}(\pi C)$ becomes 0.5

$(1 - |\vec{u}^\dagger \vec{v}|^2)$, where $|\vec{u}^\dagger \vec{v}|$ depends the angle between $\vec{u}$ and $\vec{v}$ which are unit vectors representing (pure) quantum states.

**Evaluation of the loss function.** The generator produces a density operator $Q$. The loss function is evaluated by

approximating terms like $\mathrm{Tr}(Q\psi)$ (lines 221-246) via measuring multiple copies of $Q$ (via multi-run of the generator).

**Comments on the evaluation and experiments:** Most existing literature is not explicit in architecture, with no publicly

available code/data, and has only studied the 1-qubit case (except for Ref. [3] with 6-qubit). We are the only one with a

thorough numerical study up to 8 qubits, with both large generator circuits and noisy simulation. Note that the sample

space for 8-qubit is already of dimension $2^8 \times 2^8$. This exponential growth limits numerical evaluation by classical

simulation in quantum computing and we did reach the limit of our computing resources. Our advantage to all existing

literature (especially to Ref. [3]) is demonstrated in lines 279-295. Our to-add real-word application (in response to

Reviewer #2) further demonstrates the ability of qWGAN to handle complicated tasks.

**In the revised version of the paper, we will address all minor comments and also add a background section on**

**quantum information to make our results further accessible to broader audience.**

[Meta-Review · NeurIPS 2019]

Please take into account the new comments brought forward by the new Reviewer. This accept decision is somewhat conditional on the fact that you will include more clearly these references in the final version of the paper. We strongly urge you to do so, and trust you on this, because at this point, without these references and a more clear discussion of what has been considered by other authors, the paper in its current form would be a borderline reject. Please spend at least 1/2 a page clarifying connections with prior quantum W work.